**Predicting Soil Salinity in the Red River Delta (Vietnam) Using Machine Learning and Assessing**

**Farmers' Adaptive Capacity**

Huu Duy Nguyen[1], Dinh Kha Dang[2], Thi Anh Tam Lai[1], Duc Dung Tran[3], Himan Shahabi[4], Quang-

Thanh Bui[1]

[1] Faculty of Geography, VNU University of Science, Vietnam National University, Ha Noi, 334 Nguyen
Trai, Thanh Xuan district, Hanoi City, Vietnam; nguyenhuuduy@hus.edu.vn;
laithianhtam_t65@hus.edu.vn; thanhbq@vnu.edu.vn

[2] Faculty of Hydrology, Meteorology, and Oceanography, VNU University of Science, Vietnam National
University, Ha Noi, 334 Nguyen Trai, Thanh Xuan district, Hanoi, Vietnam; dangdinhkha@hus.edu.vn

[3] National Institute of Education, Nanyang Technological University, Singapore, Singapore; [4] Earth
Observatory of Singapore and Asian School of the Environment, Nanyang Technological University,
Singapore, Singapore; [5] Center of Water Management and Climate Change, Institute for Environment and
Resources, Vietnam National University, Ho Chi Minh City, Viet Nam; dungtranducvn@yahoo.com

[4] Departments of Geomorphology, Faculty of Natural Resources, University of Kurdistan,
Sanandaj City, Kurdistan Province, Iran; h.shahabi@uok.ac.ir

Corresponding: Huu Duy Nguyen (nguyenhuuduy@hus.edu.vn)

**Abstract**

Soil salinity is a grave environmental threat to agricultural development and food security in large parts of the world,
especially in the situation of global warming and sea level rise. Reliable information on the adaptive capacity of farms
plays a key role in reducing the socioeconomic effects of soil salinization and helps policymakers and farmers propose
more appropriate measures to combat the phenomenon. The research aims to design a theoretical framework to assess
soil salinity in the Red River Delta (Vietnam) based on machine learning, optimization algorithms (namely, Xgboost
(XGB), XGB-Pelican Optimization Algorithm (POA), XGB-Siberian Tiger Optimization (STO), XGB-Serval
Optimization Algorithm (SOA), XGB-Particle Swarm Optimization (PSO), and XGB-Grasshopper Optimization
Algorithm (GOA)), remote sensing, and interviews with local people. We evaluated the geographical distribution of
soil salinity by applying machine learning to Sentinel 1 and 2A. The adaptive capacity of farmers was evaluated
through interviews with 87 households. The statistical indices, namely the mean absolute error (MAE), the root mean
square error (RMSE), and the correlation coefficient ($R^2$), were used to assess the machine learning models. The
outcome of this study demonstrated that all optimization algorithms were successful in improving the accuracy of the

XGB model. The XGB-POA had the most performance, with an $R^2$ value of 0.968, followed by XGB-STO (R²=0.967),
XGB-SOA (R²=0.966), XGB-PSO ($R^2$ = 0.964), and XGB-GOA (R²=0.964), respectively. The soil salinity map
produced by the proposed models also indicated that the coastal and riverside regions were the most affected by soil
salinity. The results also showed human and financial resources to be the two most important factors influencing the
adaptive capacity of farmers. This study provides a key theoretical framework that enhances previous previous and
can assist policymarkers and farmers in managing land resource, such as accurately identifying areas affected by soil
salinity for agricultural development in the context of climate change. In addition, this research highlights the
importance of integrating machine learning, remote sensing, and socio-economic surveys in soil salinity management,
which can support farmers for sustainable agricultural development.
**Keywords:** Red river, soil salinity, machine learning, adaptive capacity
**1. Introduction**
Soil salinity is among the greatest threats to land management, posing significant problems to agricultural progress
and global food security (He et al., 2024; Jia et al., 2024; Xiao et al., 2024). According to FAO, soil salinity affects
about 424 million hectares of land surface (with a depth of 0-30 cm) and more than 833 million hectares of subsoil
(30-100 cm). This area is increasing by about 2 million hectares each year and influences more than 100 countries
worldwide, causing damage between 12 and 27.3 billion USD (Aksoy et al., 2024; Jia et al., 2024).
The soil salinity problem will occur at the local, regional, and global levels (Bandak et al., 2024; Liu et al., 2024). In
Vietnam, many littoral regions are affected by soil salinity problems. According to the 2021 Ministry of Agriculture
and Environmental Report on the Current Situation and Planning of Agricultural Development, in 2020, about 200,000
hectares of cropland in Vietnam were already affected by soil salinity. This problem is increasingly serious in Mekong
Delta and Red River Delta - home to over 40 million people and playing a key role in Vietnam's agricultural and
aquaculture sectors - where they account for 71% of paddy cultivation, 86% of aquatic farming, and 65% of fruit
production (General Statistics Office, 2024; Ministry of Aquaculture, Agriculture and Rural Development, 2013).
Because these low-lying coastal areas (Hung and Larson, 2014) are experiencing subsidence (Le Dang et al., 2014),
and declining river water levels,, they have become highly susceptible to the effects of climate variability and sea-
level rise (Dasgupta et al., 2009). Therefore, monitoring soil salinity is essential to inform agricultural management
strategies to ensure food security at local and regional levels.

In order to address the problem, it is important to have the most precise and current data on soil salinity. Traditionally, direct field measurements of soil salinity yielded the most accurate data (Eldeiry et al., 2008; Rhoades and Ingvalson, 1971). This method collects point samples in the areas of interest one by one, which is time-consuming and requires significant manual work. Although this method can accurately identify soil salinity, it requires many field missions to collect the data over time, which can be time-consuming and resource-intensive. To reduce these limitations and obtain continuous spatial data (such as raster data) suitable for GIS analysis and environmental monitoring, several studies have used freely available remote sensing data, such as Landsat and Sentinel images. These data provide spatial (10m) and temporal (3 to 5 days) resolution and capture multiple spectral bands (Asfaw et al., 2018; Cullu, 2003). Several studies have demonstrated the effectiveness of remote sensing to monitor soil salinity accurately and rapidly. By constructing correlations between remote sensing-derived indices and soil salinity points in the field, such as NDVI, VSSI, and NDSI, we can achieve this. Although remote sensing can monitor soil salinity using different spectral responses, slightly or moderately saline soils cannot be distinguished easily because soil minerals and their components modify the spectral capacity of the soil surface.

Recently, with improvements in computing power, machine learning, and deep learning, there has beensubstantial growth in techniques to construct soil salinity maps with higher accuracy. Algorithms such as random forest (Fathizad et al., 2020), XGBoost (Jia et al., 2024), support vector machines (Jiang et al., 2019), CatBoost (Gong et al., 2023; Wang et al., 2022), and AdaBoost (Wang et al., 2022) are the most popular algorithms to construct soil salinity maps by integrating satellite images and in situ measurements. Some research has used deep learning models to construct soil salinity maps, such as deep neural networks, recurrent neural networks, and Deep Boltzmann machines. Kaplan et al. (2023) used four machine learning algorithms, namely M5P, RF, Linear, and IBK, integrated with Sentinel 2A data and 393 soil samples collected in situ to construct a soil salinity map for the United Arab Emirates. The study's results indicated that all models' performed well in assessing soil salinity, with the IBK model demonstrating the highest effectiveness. Aksoy et al. (2024) used XGBoost and random forest with 26 environmental covariates from Landsat 8 OLI to evaluate soil salinity in Iran's Lake Urmia. The study's outcome showed that machine learning integrated with Landsat 8 OLI data successfully monitored soil salinity, with XGB yielding more accurate results than random forest. Jia et al. (2024) applied nine models, namely PLSR, Lasso, CART, RF, ERT, GBDT, LightGBM, XGBoost, and AdaBoost, integrated with Sentinel 2A imagery, to evaluate soil salinity in the Ningxia Yellow River Diversion Irrigation Area. The results showed that the AdaBoost model performed better than the others.

Previous studies show that although machine learning methods have been utilized to assess soil salinity in many
regions of the world, their application for this purpose is still limited in the Mekong and Red River Deltas (Shi et al.,
2021; Vermeulen and Van Niekerk, 2017). Currently, there are only four studies that have assessed soil salinity in the
Mekong Delta (Hoa et al., 2019; Nguyen et al., 2023; Nguyen et al., 2018; Nguyen et al., 2021), and no work has been
done in this field for the Red River Delta. In addition, most previous studies have developed state-of-the-art methods,
such as integrating machine learning and remote sensing, to identify the geographical distribution of soil salinity in
different regions of the world (Hardie and Doyle, 2012; Su et al., 2020; Wang et al., 2007). While several studies have
highlighted the importance of assessing the adaptation capacity of the community to strengthen their resilience to soil
salinity and other types of natural hazards, however, very few studies integrate this aspect into the identification of
spatial soil salinity..Hoang et al. (2023) reported that assessing the ability of farms to adapt to soil salinization is the
key to reducing vulnerability and contributes significantly to the development of sustainable livelihoods.
The adaptive capacity is defined as the capability of the community to cope, adjust, and adapt to the impacts of growing
soil salinity. It measures the ability to predict, respond, and recover from the phenomenon. It is assessed on different
scales, using different approaches, according to the region in question (Mazumder and Kabir, 2022; Thiam et al.,
2024). Furthermore, understanding the adaptive capacity of communities plays an important role in reducing the
negative effects of salinity intrusion in coastal regions in general and the Red River Delta in particular. By assessing
adaptation at multiple scales with site-specific methods, researchers and local governments can identify interventions
(such as crop variety changes, crop calendars, irrigation systems) that are effective. The IPCC in 2014 indicated that
farm adaptive capacity depends on five main factors: natural capital, human capital, material resources, financial
resources, and social capital. Therefore, integrating the adaptive capacity of populations with the soil salinity map
improves the accuracy of predictions and proposes adaptation strategies that strengthen the overall resilience of
communities.
The research aims to improve a theoretical framework to assess soil salinity and farmers' adaptive capacity based on
machine learning, optimization algorithms (namely XGB, XGB- POA, XGB- STO, XGB- SOA, XGB- PSO, and
XGB- GOA), remote sensing, and interviews with local people. Several studies have examined farmer's adaptive
capacity to environmental stressors in different regions (Bhuyan et al., 2024; Thiam et al., 2024). However, no studies
comprehensively analyze farmers' adaptive ability to combat soil salinity in a given region based on machine learning,
remote sensing, and interviews with local people. In addition, several studies combine machine learning with Sentinel
1 or Sentinel 2 to assess soil salinity (Wang et al., 2021; Xiao et al., 2023); however, there are rarely studies that
combine machine learning with Sentinel 1 and Sentinel 2 to monitor soil salinity in the Red River Delta. By the
combination of Sentinel 1 and Sentinel 2, advanced machine learning, and the information from farmers themselves,
this study filled a critical gap and provided a novel, comprehensive framework for monitoring and responding to soil
salinity in the Red River Delta (Ma et al., 2021).
In general, salinity intrusion harms agricultural development and people's livelihoods. Therefore, it is necessary to
develop a theoretical framework to address the soil salinity problem in terms of natural and social factors. However,
previous studies have mainly assessed the spatial distribution of salinity or the community's adaptive capacity, and
hardly any studies have assessed both the spatial distribution of salinity and the community's adaptive capacity. Thus,
the global contribution of this study is to fill the knowledge gap about the spatial distribution of soil salinity and the
adaptive capacity of communities in the Red River Delta in general and Thai Binh Province in particular by relying
on modern methods to improve this important and understudied understanding. The results of this study can play an
important role in mitigating the impact of salinity intrusion on agricultural development and can help policymakers
and planners develop effective strategies to mitigate this impact, especially in the context of climate change.
**2. Study Area**
The Red River Basin covers a total area of 169,000 km² and spans China (48%), Laos (0.7%), and Vietnam (51.3%).
The river system has a total length of 1,150 km, with around 500 km in the territory of Vietnam before discharging
into the Gulf of Tonkin. The topography is mainly mountainous terrain that comprises about 70% of the total area at
elevations above 500 meters. In the lowlands, elevations range from approximately 0.4 to 9 m, characterized by a
tropical climate with summer monsoons from the south and winter monsoons from the northeast (Vinh et al., 2014),
the basin experiences average annual precipitation ranging from 800 to 3000 mm. The rainy season occurs from May
to October and accounts for 70%-90% of annual rainfall (Quang et al., 2024). Daily rainfall varies from 300-400 mm
during this period. The average temperature ranges from 22 to 27 °C, with winter temperatures potentially below 10
°C and summer temperatures above up to 40 °C.
The basin flows into the Gulf of Tonkin through nine river mouths, of which the Tra Ly, Van Uc, and Ba Lat are the
main channels for water conveyance. These channels transport a substantial sediment load of approximately $120 \times 10^6$
tons annually to the Red River Delta region (Vinh et al., 2014). The littoral region has a semi-diurnal tidal regime,
with tidal ranges ranging from 2 to 4 m. Saline intrusion significantly influences the littoral region during the dry
season with average and maximum wave heights of about 0.7-1.3 m and 3.5-4.5 m, respectively. However, during
major storms, wave heights can reach 5 m (Nhuan et al., 2007).
The Red River Delta is influenced by several natural hazards, such as flooding, soil salinity, and sea level rise
(Castelletti et al., 2012).  Several studies have highlighted that rising sea levels are having an increasingly severe
impact on inland regions, leading to soil salinity (Nguyễn Văn Đào, 2023). Recently, the Red River Delta in general
and the Thai Thuy district in particular have been affected by soil salinity, causing significant damage to agricultural
development and negatively impacting residents' livelihoods (Figure 1).

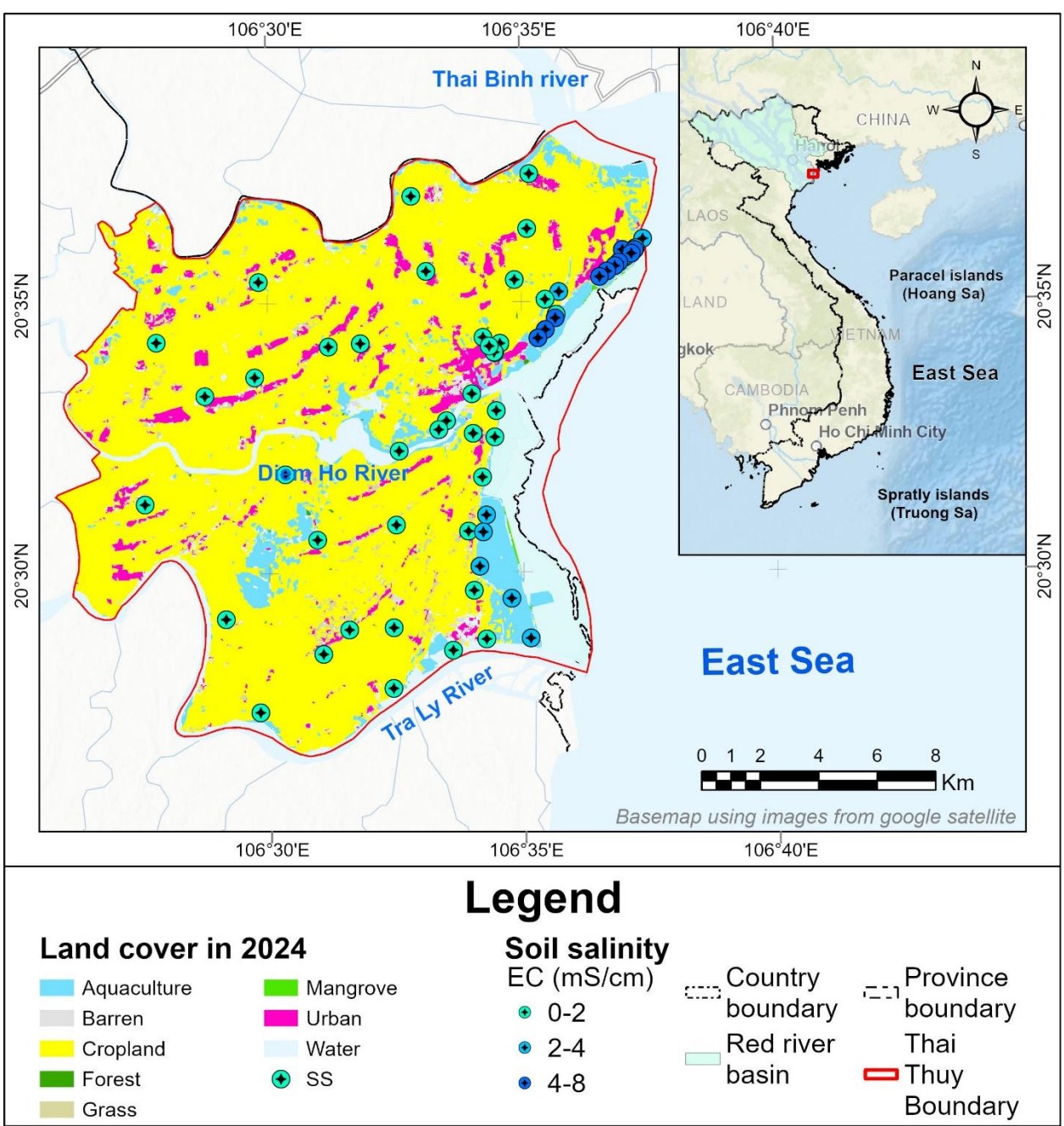


Figure 1: Geographical location of study area: The red boundary on the map represents the Thai Thuy district, located in the Thai Binh province in the Red River Delta of Vietnam. The green points are the soil salinity samples collected in April 2024. The land use in the Thai Thuy district is divided into eight types: aquaculture, barren land, cropland, forest, grass, mangrove, urban, and water body. While the aquaculture area is located in the coastal zone, cropland takes up a large part of the study area, and Urban is located in the center and along the road.

155

**3. Methodology**

The first strand of the methodology was the identification of the soil salinity mapping. We divided this process into four main steps (Figure 2):

Preparation of soil salinity samples and factors

 The data for constructing the soil salinity map were divided into two main types: EC and conditioning factors.

*EC Measurements*

According to the FAO report, if the sea level rises by 50 cm, about 11.8% of coastal land is at risk of being flooded by salt water; this figure increases to about 31.4% if the sea level rises by 100 cm in Thai Binh province. Recently, saltwater intrusion has clearly affected agricultural production in Thai Binh province and Thai Thuy district. Typically, in the 2015-2016 winter-spring crop, the salinity in the main rivers exceeded the threshold of 1‰, causing significant damage to crops. In 2020, the salinity in the main rivers, 28 km from the sea, exceeded the threshold of 3.75‰, far exceeding the allowable threshold of 2.75‰. This phenomenon greatly affects agricultural production in the area, underlining the urgent need for systematic soil salinity assessment.

To address this issue, we collected soil salinity samples using soil drills, applying both zigzag and grid sampling techniques, which are frequently employed in small-scale studies (Elshewy et al., 2024; Jia et al., 2024). The sampling depth depends on the soil salinity assessment for each specific crop. This study monitors soil salinity with the objective of agricultural development; therefore, soil samples were obtained from a depth of 0 to 30 cm. The sampling process occurred in the dry season, between March and April 2024. In addition, when sampling in the field, it is necessary to consider the homogeneity of the soil. We collected 62 samples to cover the entire field. We collected samples along the road to identify different types of soil, and the farmers labeled the samples accordingly. The samples were locked in bags until analysis in the laboratory. We noted the positions of the samples, including longitude and latitude, during the sampling process. When the samples were sent to labs, they were stored in enameled jars, and impurities like stones, wood, and branches were removed. These soil samples were then finely ground. The electrical conductivity (EC) was then calculated from a 1:5 soil/deionised water suspension. A soil/water suspension was created by adding 7 g of soil to 35 ml of distilled water and then mixed with a mechanical stirrer for 60 minutes to dissolve the salt. The

EC value was measured using a conductivity probe. Finally, the samples were split into two parts: the first part (70%)
was used to build the proposed models, while the other part (30%) was applied to confirm the model.
*Remote Sensing Data*
Due to the effects of the earth's cycle, the salt accumulated in the soil is closely linked to climatic conditions,
hydrology, soil characteristics, and surface vegetation density, for example, topographic characteristics (Wang et al.,
2024; Xie et al.). We calculated these factors using optical Sentinel 2A images and microwave Sentinel 1 images to
determine the soil salinity value. The Sentinel-2A images were calculated by running Sen2Cor for atmospheric
correction to ensure the transition between apparent atmospheric reflectance and surface reflectance, and these images
were obtained using Google Earth Engine. To reduce the influence of clouds, Sentinel 2A images for March-April
2024 were selected with a cloud cover rate of less than 10%. To enhance the quality of these images, the median value
of each pixel was calculated at a resolution of 10 m. As for the Sentinel 1 images, they were acquired in dual cross-
polarization mode and broadband interferometric mode. The median value of the Sentinel 1 image obtained on March
31, 2024, was computed to acquire microwave remote sensing data at a scale of 10 m. As well as 12 bands of the
Sentinel 2A image (from band 1 to band 12) and 3 indices of the Sentinel 1 image (VV, VH, and VVVH). In addition,
20 spectral indices extracted from Sentinel 2A image were selected to build the soil salinity model, namely Brightness
index (BI), Canopy Response Salinity Index (CRSI), Enhanced Vegetation Index (EVI), Intensity index 1 (Int1),
Indensity index 2 (Int2), Normalized Difference Salinity Index (NDSI), Normalized Difference Vegetation Index
(NDVI), Ratio Vegetation Index (RVI), Salinity index (S1), Salinity index (S2), Salinity index (S3), Salinity index
(S5), Salinity index (S6), Soil Adjusted Vegetation Index (SAVI), Salinity Index 1 (SI), Salinity index 2 (SI1), Salinity
index 3 (SI2), Salinity index 4 (SI3) and Salinity index 5 (SI4). These factors were divided into three main groups:
vegetation indices (NDVI, CRSI, RVI, SAVI, and EVI), water indices (flow direction and distance to river), salinity
indices (SI, SI1, SI2, SI3, SI4, S1, S2, S3, S5, S6, and NDSI), topography indices (elevation and slope), and
chlorophyll spectral indices (BI, Int1, Int2). These indices have been used frequently in previous studies (Hoa et al.,
2019; Nguyen et al., 2021; Wang et al., 2021).
The indices due to the vegetation reflect the health and growth of vegetation, and the present study indirectly reflects
the level of soil salinity in any region. The increase in soil salinity has a negative effect on the development of
vegetation due to the difference in the absorption of water and nutrients; therefore, it leads to a decrease in the values
of NDVI, RVI, SAVI, and EVI and an increase in the value of CRSI (Jia et al., 2024; Wang et al., 2024). Water indices
play an important role because regions near rivers or along the flow path are more affected by the salinity risk. River
flow is often affected by tides or seawater intrusion; therefore, when the distance to the river decreases, the salinity
risk increases due to the infiltration of salty river water into the soil. Furthermore, the flow direction influences the
propagation and infiltration of water in the soil. Salty water can penetrate further inland if the flow is from the sea to
the river, especially in the dry season (Nguyen et al., 2021).
Topography indices are key in constructing soil salinity models, as salty water penetrates low-lying regions more
easily. In the Red River Delta, the low-lying regions are located along the coastline, and as such, these regions are
more affected by the risk of soil salinity. Salinity indices highlight the value of spectral reflectance in regions affected
by saltwater intrusion. The higher the salinity, the higher the spectral reflectance value (Du et al., 2024).
Construction of hybrid machine learning models
We built six machine-learning models to identify the spatial distribution of soil salinity. This involved two main steps:
constructing an individual XGB model and then creating hybrid models by integrating each of five optimization
algorithms with the XGB model. We developed the XGB model using Python and the Sklearn library.
The machine learning model-building process was divided into two main steps: the first was the XGB model building,
and the second was the hybrid model building (the integration of XGB with optimization algorithms). The accuracy
of the machine learning model depends on the parameter adjustments of the XGB model. In this study, the XGB model
parameters were selected using the trial-and-error method. Finally, the XGB parameters were n_estimators=100,
max_depth=4, subsample=0.5, and colsample_bytree=0.5. While the hybrid model was built by integrating the XGB
model and optimization algorithms, namely GOA, POA, SOA, STO, and PSO. To integrate the XGB model with
optimization algorithms, we first need to construct an objective function $F(\theta)$ that returns the error value of XGB on
the validation set when using the parameter sets $\theta$. That is, each parameter set has a different error value. Next,
determine the search space of the hyperparameters (n_estimators, max_depth, subsample, colsample_bytree) as
discrete value intervals. Then, the optimization algorithms will initialize the population of individuals with the size
and parameters characteristic of each optimization algorithm. This study was tested with 500 iterations: at each
iteration, each individual will generate a combination of $\theta i$, and the optimization algorithms will update the position
or velocity of the individuals according to their own rules. This process is repeated until a stopping threshold is set.
Finally, the results are the optimal parameters. The parameters of the model are as follows: problem_size =
3, batch_size = 25, epoch = 500, pop_size = 50, "fit_func": fun_avr2, "lb": [0] *problem_size, "ub": [1]* problem_size,
c_min = 0.00004, c_max = 2.0 for **XGB-GOA**; problem_size = 3, batch_size = 25, epoch = 500, pop_size = 50,
"fit_func": fun_avr2, "lb": [0] *problem_size, "ub": [1]* problem_size, c1=2.05, c2=2.05, w_min=0.4 for **XGB-PSO** ;
problem_size = 3, batch_size = 25, epoch = 500, pop_size = 50, "fit_func": fun_avr2, "lb": [0] *problem_size, "ub":*
*[1]* problem_size  for **XGB-POA**; problem_size = 3, batch_size = 25, epoch = 500, pop_size = 50, "fit_func":
fun_avr2, "lb": [0] *problem_size, "ub": [1]* problem_size for **XGB-SOA**; problem_size = 3, batch_size = 25, epoch =
500, pop_size = 50, "fit_func": fun_avr2, "lb": [0] *problem_size, "ub": [1]* problem_size for **XGB-STO**.
Evaluation of model accuracy
The statistical indices, the root mean square error (RMSE), the mean absolute error (MAE), and the correlation
coefficient (r) were used to assess the accuracy of the proposed models.
Analysis of spatial distribution and identifying the farmers adaptive capacity
We validated the models and then used them to assess soil salinity in the study area at a pixel scale with a resolution
of 10x10 m. Approximately 30 million pixels were assessed, and a soil salinity map was constructed using the Point
to Raster tool in ArcGIS 10.8.
We used the second strand of our methodology, interviews with populations, to evaluate the adaptive capacity of farms
in the study area. We selected An Tan commune in the Thai Thuy district to participate in structured interviews. A
total of 87 households were interviewed. These households were randomly selected from An Tan commune in the
Thai Thuy district to participate in structured interviews. The commune is located in the coastal region, which has the
lowest altitude and often affected by soil salinity. Residents mainly worked in rice and fish farming. We analyzed all
87 responses to assess farmers' ability to adapt to soil salinity.
The structured interviews focused on five main elements: natural resources, human resources, physical resources,
financial resources, and social resources. There was a particular focus on soil salinity in 2023, allows to evaluate the
stress of environment on the livelihood of population.

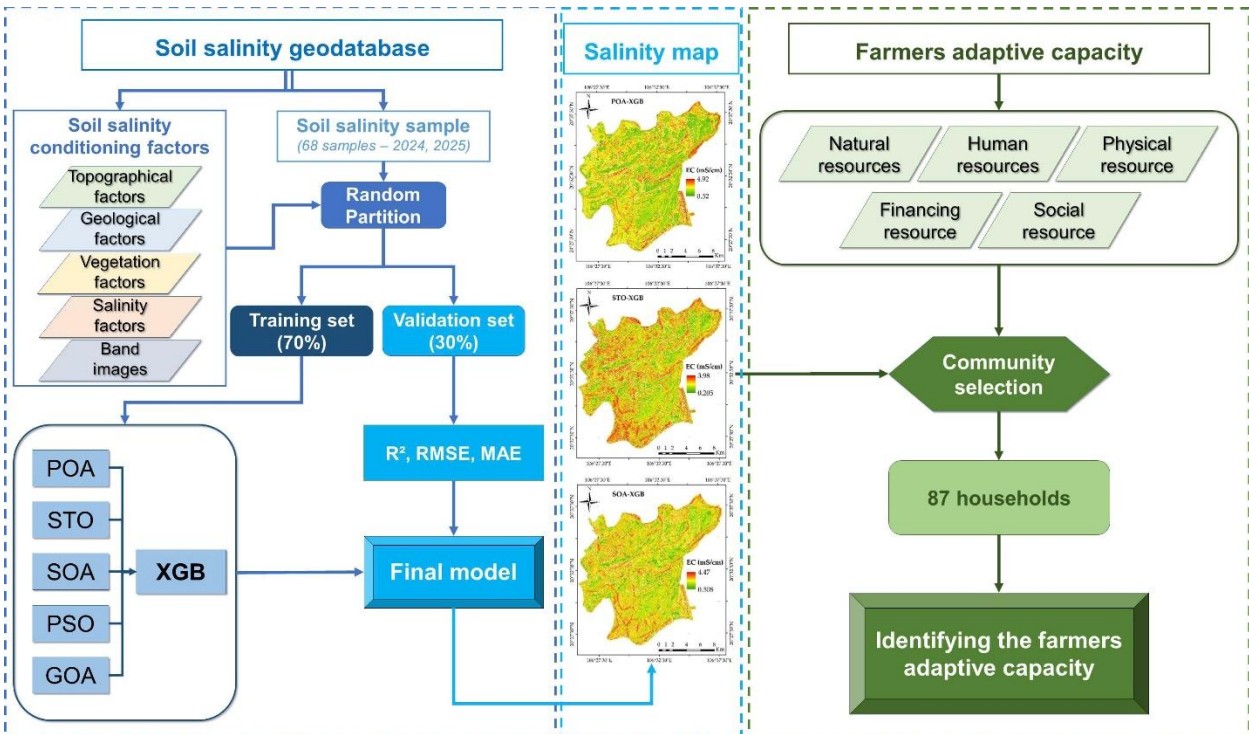


Figure 2: Methodology used for the farmer's adaptive capacity and soil salinity in this study: The methodology used
in this study is to design a theoretical framework to assess soil salinity and farmers' adaptive capacity based on
machine learning, optimization algorithms, remote sensing, and interviews with local people. We divided this
process into four main steps: i) data preparation; ii) machine learning model construction; iii) machine learning
model evaluation; and iv) analysis of spatial distribution and identifying the farmers' adaptive capacity.

## 3.1. XGBoost (XGB)

XGB is a popular gradient-boosted tree algorithm that can solve classification and regression problems. The main idea
of learning with XGBoost is to train several models sequentially and combine them successively by correcting errors
iteration after iteration to obtain the most potent ensemble model possible (Zhang et al., 2022). Therefore, the
prediction result consists of a set of chained decision trees. This method increases the performance and stability of the
model while minimizing its variance (Zhang et al., 2022). The XGB model functions in three main steps: i) it
constructs an individual model (tree) by taking predictions on the training data, ii) it computes the mistakes of these

predictions for the real values, and iii) it constructs another tree to predict and correct these mistakes. The process is
repeated, and each new tree is built to correct the mistakes of the previous one. This phase is called "boosting". We
then sum up the predictions of all trees to determine the final predictions (Mukhamediev et al., 2023).
To optimize the accuracy of the XGB model, three main parameters need to be adjusted: learning rate (reducing the
value of this parameter can avoid the overfitting problem), alpha, and lambda (increasing the value of these parameters
makes the model more conservative), and column sample by tree (adjusting this parameter has the objective of
obtaining the subsample of columns) (Tan et al., 2023).
**3.2. Pelican Optimization Algorithm (POA)**
Agents searching for prey in nature have a mechanism similar to that of agents searching for optimal solutions.
Therefore, based on this perspective, the search agents that comprise a population seek to achieve the optimal solution
more quickly. Each agent is an optimal solution whose position is determined in the search space. From a mathematical
perspective, agents are vectors, and the population of agents forms matrices (Dehghani and Trojovský, 2022). Among
the values used to calculate the aim function, the top value of the agents is determined as the top solution of the agents.
One such optimization algorithm is POA proposed by Trojovský and Dehghani (2022). This algorithm is designed
based on Pelicans' foraging and communication processes. This is a large bird with a long beak and a large pouch in
the throat to hold prey during hunting. Hundreds of individuals may flock together. They can weigh up to 15 kg, grow
to a height of 1.8 m, and have a wingspan of up to 3 m. These characteristics greatly assist them in finding food, such
as fish, frogs, and turtles.
POA is based on the simulation of the behavior and plan of pelicans when attacking prey. Pelican hunting strategies
are divided into two stages. First, the bird moves towards its prey, then it spreads its wings and glides along the water
surface to attack. In the first stage, the pelicans determine the situation of the prey and move toward the identified
prey area. Identifying prey areas represents the determination of the search space in the POA model. The positions of
the prey are randomly produced in the search space, which increases the exploration power of POA in the process of
searching and solving optimization problems. After locating the prey area in the second stage, pelicans spread their
wings and move on the water surface to attack the prey and store it in their throat pouch. This strategy allows them to
capture more prey. Modelling this behavior of pelicans makes the POA model easier to converge and improves local
search ability (Alamir et al., 2023; Trojovský and Dehghani, 2022).

**3.3. Serval Optimization Algorithm (SOA)**

SOA is one of the population-based optimization algorithms, and it exploit the searching power of agents in a
population. This property makes this algorithm powerful in solving optimization problems (Dehghani and Trojovský,
2022). Each iteration of SOA continuously determines and updates the agents' situations. The updating process
simulates the behavior of serval cats in the wild, divided into two stages: i) exploration of the search space and ii)
local exploitation in the search space (Sindi et al., 2024).
Wild cats are some of the most efficient predators, using hearing to identify and attack prey. In the first stage of SOA,
the situation of the servals is repeatedly up-to-date after each move: the continuous updating of positions leads to
detailed coverage of the search space. The this stage aims is to raise the power to search and explore in the search
space. The situation of the best member in the SOA is considered the situation of the prey and, therefore, the optimal
solution (Dehghani and Trojovský, 2022; Sindi et al., 2024).
When attacking prey, wild cats jump during the chase to prevent the prey from escaping. These strategies also serve
to update the position of the SOA population. The simulation of the chase process can cause small changes in agents'
positions in the SOA. However, this phase aims to increase the search space mining capability of SOA, which helps
to improve the local search capability in the search space (Dehghani and Trojovský, 2022).

**3.4. Siberian Tiger Optimization (STO)**

The STO algorithm is a new biologically inspired hyper-heuristic algorithm modeled after Siberian tigers' hunting
behavior (Trojovský et al., 2022). STO replicates the Siberian tiger tracking and capture strategy, using a population-
based approach to explore the search space efficiently and quickly (Trojovský et al., 2022).
STO works by simulating the way Siberian tigers move and communicate with each other while hunting their prey.
Each agent in the STO algorithm represents a Siberian tiger, exploring a different region in the search space. The
tigers communicate and share information about their locations with each other to find the optimal location. The
location update process of Siberian tigers in STO is carried out in two main phases: the hunting and bear-fighting
phases (Trojovský et al., 2022).
In the hunting phase, since the Siberian tiger is a powerful predator, it hunts by attacking different prey, so the agents
in the STO are up-to-date based on the simulation of this hunting strategy. After choosing the prey, the Siberian tiger
will chase, attack, and kill its prey in this phase. The population continuously updates the members' positions based
on the selection and attack of the prey. This phase causes sudden changes in the members' positions and increases the
search ability in the search space (Al-Sarray et al., 2024).
During hunting, the Siberian tiger has to fight with brown and black bears. Therefore, in the second phase, the members
of the STO stay update on the strategies used by the Siberian tiger when bear-fighting. When fighting with bears, the
Siberian tiger ambushes and then assaults the bear until it kills it (Al-Sarray et al., 2024; Trojovský et al., 2022).
One of the key features of STO is its ability to balance exploration and exploitation. In put it another way, the design
of the STO algorithm involves extensive exploration of the search space and refinement of promising solutions in the
most promising areas. Thus, STO avoids local optimization problems and increases the likelihood of global
optimization (Trojovský et al., 2022).
**3.5. Particle Swarm Optimization (PSO)**
PSO was proposed by Kennedy and Eberhart (1995). It is founded on the principles of self-organization that allow
one or more groups of living organisms to move together in a complex way (Fu et al., 2018). PSO simulates the
movement process of some animals, such as flocks of birds. In this model, birds move randomly by following three
rules: i) they track the same path as their friends, ii) they are enticed to the average situation of the group, and iii) they
maintain a certain space between each other to avoid collisions (Ruidas et al., 2022).
PSO explores the search space through the birds' position and flight paths. The position of each bird in the search
space is considered a potential solution. More precisely, the position and speed of the birds are represented by vectors
with D dimensions, and the initial speed is determined randomly. In the PSO algorithm, the position and velocity of
each bird are updated continuously after each iteration until an optimal solution is reached. The optimization function
assesses the position and velocity quality (Bui et al., 2016). In this study, PSO was used to optimize the XGB model.
**3.6. Grasshopper Optimization Algorithm (GOA)**
GOA was first proposed by Mirjalili et al. (2018). This algorithm is based on the swarm behavior of locusts during
foraging to solve optimization problems. Grasshoppers move quickly to explore spaces, and then they move locally
to exploit resources in the foraging space. GOA models the behavior of a virtual swarm of grasshoppers, where each
position represents an optimization solution to the problem (Moayedi et al., 2021; Nguyen, 2022). Movement is
influenced by several factors: social interaction, gravity, and wind advection. Social interaction plays an important
role in finding the optimal position because grasshoppers interact with each other to exchange information about
precise positions. This social communication allows grasshoppers to find the right solutions. Then, gravity allows
grasshoppers to explore the foraging spaces in a balanced manner, hence avoiding the local optimization problem
(Ingle and Jatoth, 2024). Finally, wind advection represents the external effects that can influence the movement of
grasshoppers, leading them to some areas of the search space. In the optimization process, an equilibrium between
exploration and exploimportants is important to accurately approach the true global optimum (Moayedi et al., 2020).
**4. Results**
**4.1. Soil Salinity Predictors**
The choice of suitable factors plays a key role when using machine learning to determine the geographical distribution
of soil salinity in any region. Conditioning factors represent the causes of soil salinity, so improper selection of these
factors can result in inaccurate prediction. Data redundancy may complicate the model and lead to poor performance.
In this study, we used RF to measure the appropriate factors. It assigns a value to each factor based on its relationship
with the soil salinity samples and the conditioning factors. The most important factor is the one with the greatest
importance in determining soil salinity zones. In addition, after using RF to determine the importance of factors, we
used trial and error to continue eliminating factors that affected the precision of the model.
The outcome showed that six factors (DEM, RVI, B2, S6, S2, and S1) had an RF value of zero, so these factors did
not affect the determination of the spatial distribution of saline areas. In addition, two factors (NDVI and flow
accumulation) were eliminated using the trial-and-error method. The other 30 factors were used to build the model.
VVVH (0.39), VV (0.3), distance from the river (0.28), CRSI (0.24), and EVI (0.21) had a strong influence on the soil
salinity in the study area. S3 (0.17), BI (0.17), B12 (0.15), SI2 (0.14), B7 (0.11), VH (0.1), and Int2 (0.1) have moderate
relationships with the soil salinity. B11 (0.08), S5 (0.06), slope (0.06), SI (0.06), SAVI (0.06), NDSI (0.06), SI4 0.05),
B5 (0.05), Int1 (0.05), B9 (0.05), B4 (0.05), SI (0.04), SI1 (0.04), B8 (0.04), B3 (0.02), B1 (0.02), and B6 (0.02) had
only a weak relationship on soil salinity (Figure 3).

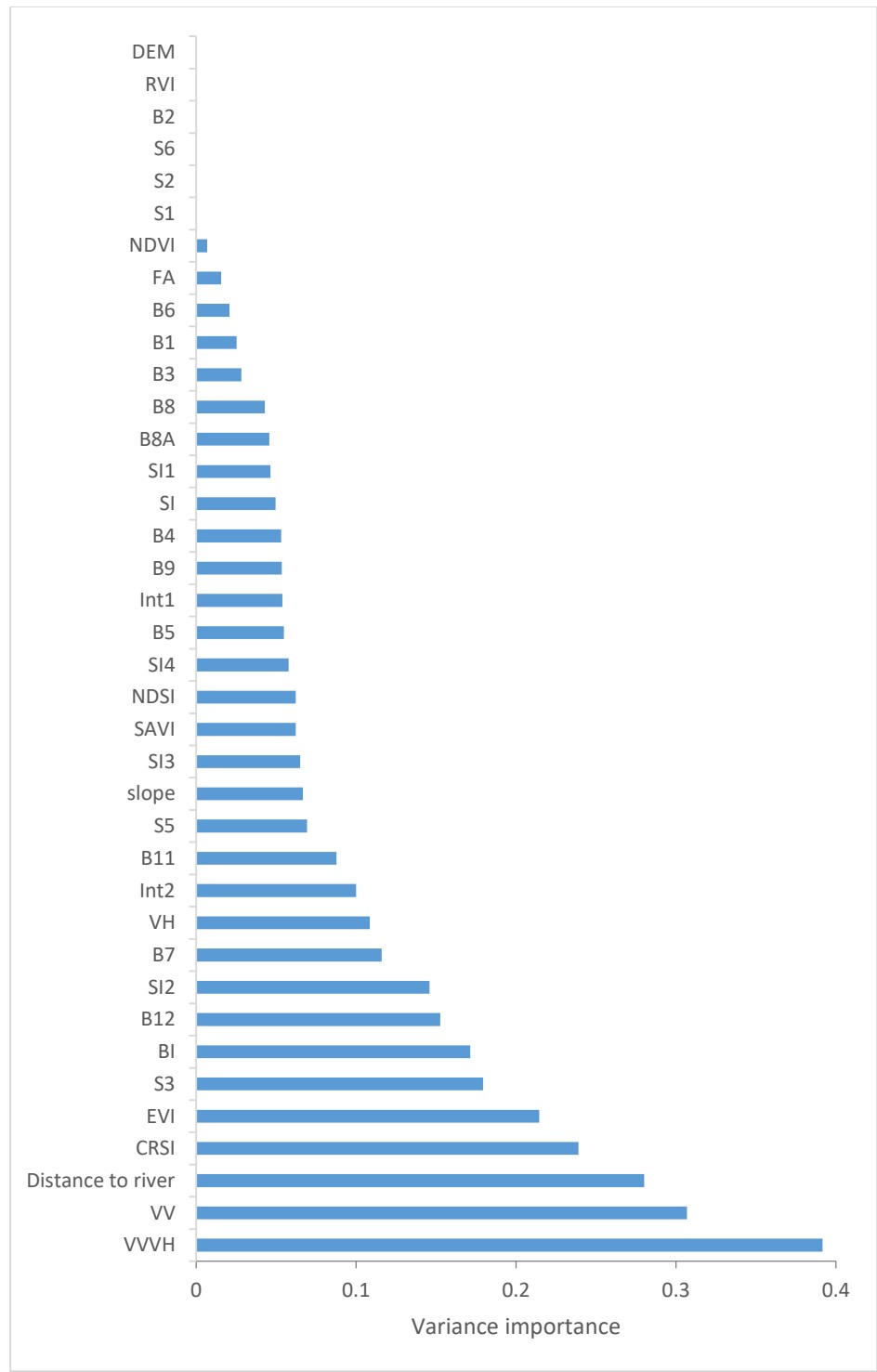

**Figure 3:** Variables important for soil salinity model using RF.

**4.2. Model Accuracy Validation**
$R^2$ was used to assess the performance of the machine learning models. The outcome of this study demonstrated that
all optimization algorithms enhanced the performance of the XGB model. The XGB-POA model was the greatest,
with an $R^2$ value of 0.968, followed by XGB-STO ($R^2$=0.967), XGB-SOA ($R^2$=0.966), XGB-PSO (R2 = 0.964), and
XGB-GOA ($R^2$=0.964; Figure 4).

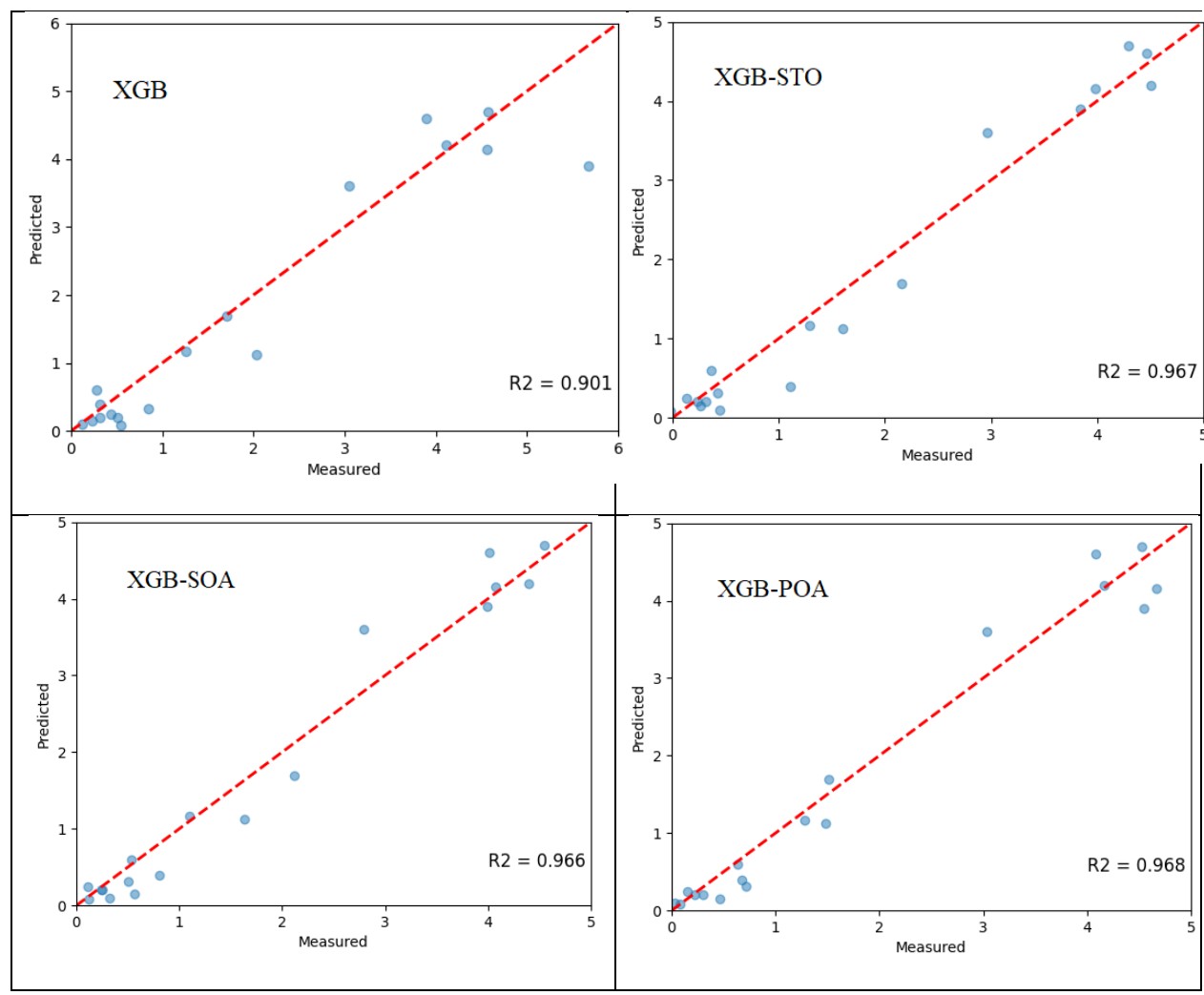

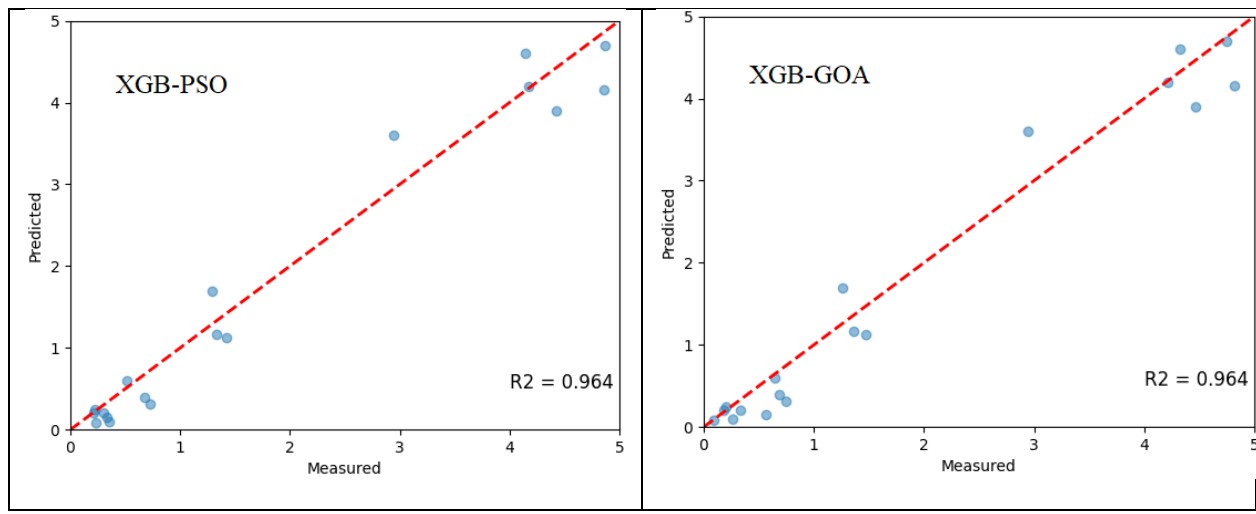

**Figure 4:** R² value for the testing dataset

The RMSE and MAE were also used to evaluate the accuracy of the machine learning models. The XGB-POA model
performed better on training and validation data (RMSE=0.28, MAE=0.18 for learning data and RMSE=0.31 and
MAE=0.242 for verification data). The XGB-STO was ranked second with an RMSE value of 0.3 and MAE of 0.22
for learning data, and an RMSE value of 0.32 and MAE of 0.244 for verification data. The XGB-SOA model was
ranked third, with RMSE=0.31 and MAE=0.23 for learning data and RMSE=0.33 and MAE=0.25 for verification
data. XGB-GOA model came fourth, with RMSE=0.33 and MAE=0.25 for learning data and RMSE=0.34 and
MAE=0.26 for verification data. The XGB-PSO model performed less well than the other models, with RMSE=0.335
and MAE=0.26 for learning data and RMSE=0.341 and MAE=0.27 for verification data (Table 1).

Table 1. Model performance and comparison.

| Models | Training dataset | | | Validation dataset | | |
|---|---|---|---|---|---|---|
| | RMSE | MAE | R² | RMSE | MAE | R² |
| **XGB-POA** | 0.28 | 0.18 | 0.99 | 0.31 | 0.242 | 0.968 |
| **XGB-STO** | 0.3 | 0.22 | 0.987 | 0.32 | 0.244 | 0.967 |
| **XGB-SOA** | 0.31 | 0.23 | 0.98 | 0.33 | 0.25 | 0.966 |
| **XGB-GOA** | 0.33 | 0.25 | 0.97 | 0.34 | 0.26 | 0.964 |
| **XGB-PSO** | 0.335 | 0.26 | 0.97 | 0.341 | 0.27 | 0.964 |
| **XGB** | 0.35 | 0.32 | 0.91 | 0.37 | 0.34 | 0.901 |


**4.3. Spatial distribution of soil salinity in the Thai Thuy district of the Red River Delta**

Following validation, we constructed a geographical distribution map of soil salinity using the proposed models. We carried out the process by assigning conditioning factors to the 30 million pixels across the entire study area. Depending on each model, the EC value varies from 0.29 to 7.7 mS/cm. On the map, the color varies from green to red, representing different EC values. The areas with green colors are located far from the continent (EC = 0.29), while the areas with red colors are located on the coast, with EC values superior to 7.7 mS/cm. This graph shows that these areas are directly affected by saltwater intrusion from the sea (Figure 5).

According to the FAO, salinity can be divided into 5 levels: non-saline, slightly saline, moderately saline, heavily saline, and very heavily saline. In which the EC value is below 2 mS/cm, the soil is considered not saline, and the plants grow completely normally. The EC value ranges from 2- 4 mS/cm; the soil is slightly saline and has very little effect on the plants. Specifically, flower crops may grow slowly, while rice and fruit trees may have reduced height. The EC value ranges from 4-8 mS/cm; the soil is moderately saline and reduces crop yields. Specifically, rice can have a 10-20% reduction in yield. The EC value ranges from 8 - 16 mS/cm; the soil is heavily saline and affects the growth of plants. The soil becomes extremely saline and uncultivated if the EC value surpasses the threshold of 16 mS/cm. This study utilizes the XGB-POA model, which boasts the highest accuracy, to analyze the areas affected by saline intrusion. Specifically, in the study area, about 65 km² of land area is not affected by saline intrusion, 165 km² is slightly affected, and 1.8 km² is greatly affected by saline intrusion. Compared to the land cover/land use map, it can be seen that the land area greatly affected by saline intrusion is mainly aquaculture; therefore, this area will not be significantly affected in terms of productivity. Meanwhile, 165 km2 of land affected by saline intrusion is rice land, which can slow down rice growth and reduce productivity.

In the study area of Thai Thuy District, Thai Binh Province, about 70-75% of the agricultural land is irrigated by a canal system that draws water from the Red River and the Day River. Thanks to the water source from the Red River and the Day River, agricultural production areas are regularly washed away with salt, so the EC value in these areas is often below 4 mS/cm. In coastal areas, salinity intrusion at river mouths forces people to use shallow groundwater for irrigation, which leads to salt accumulation and prompts a shift from rice cultivation to aquaculture.

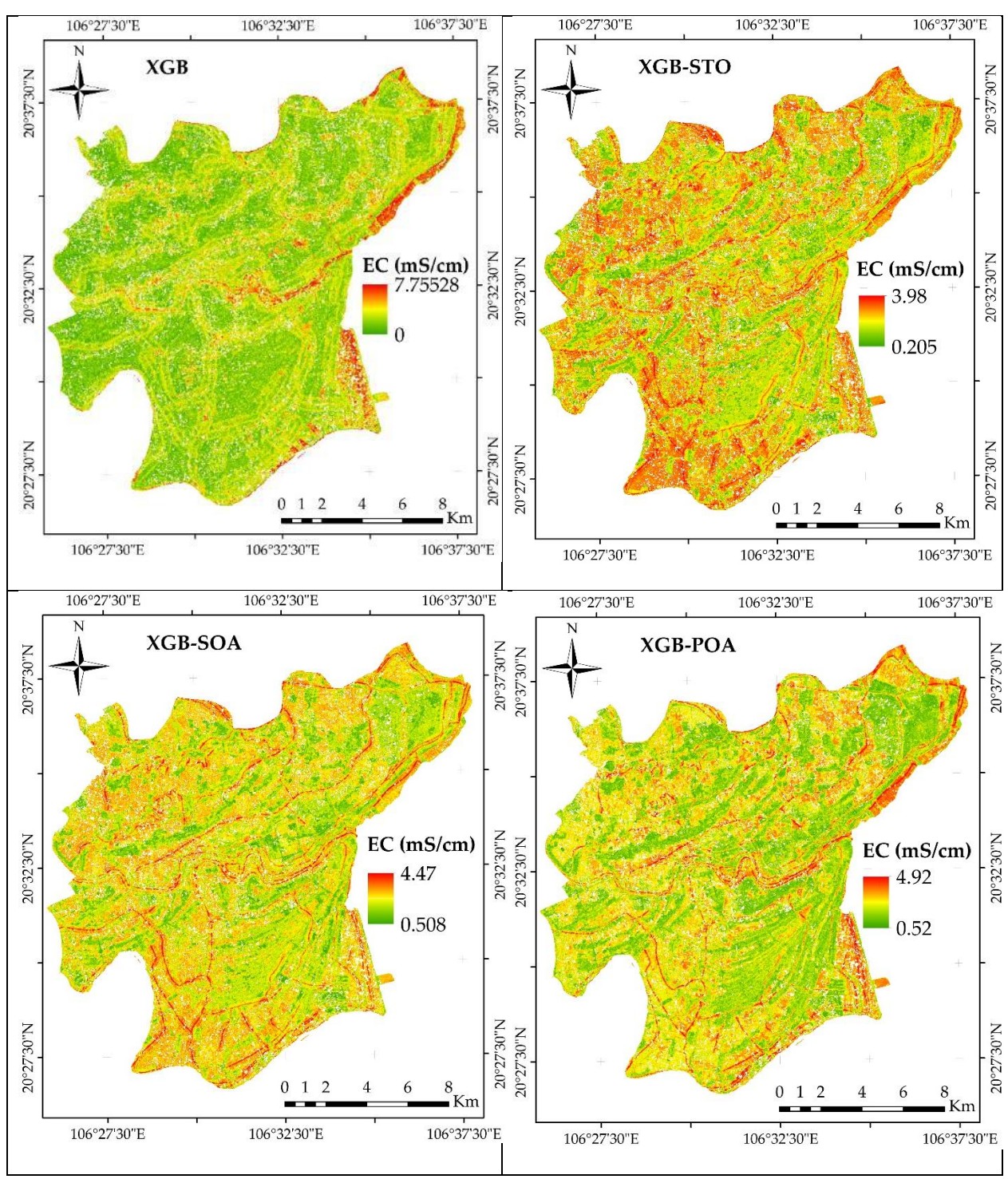

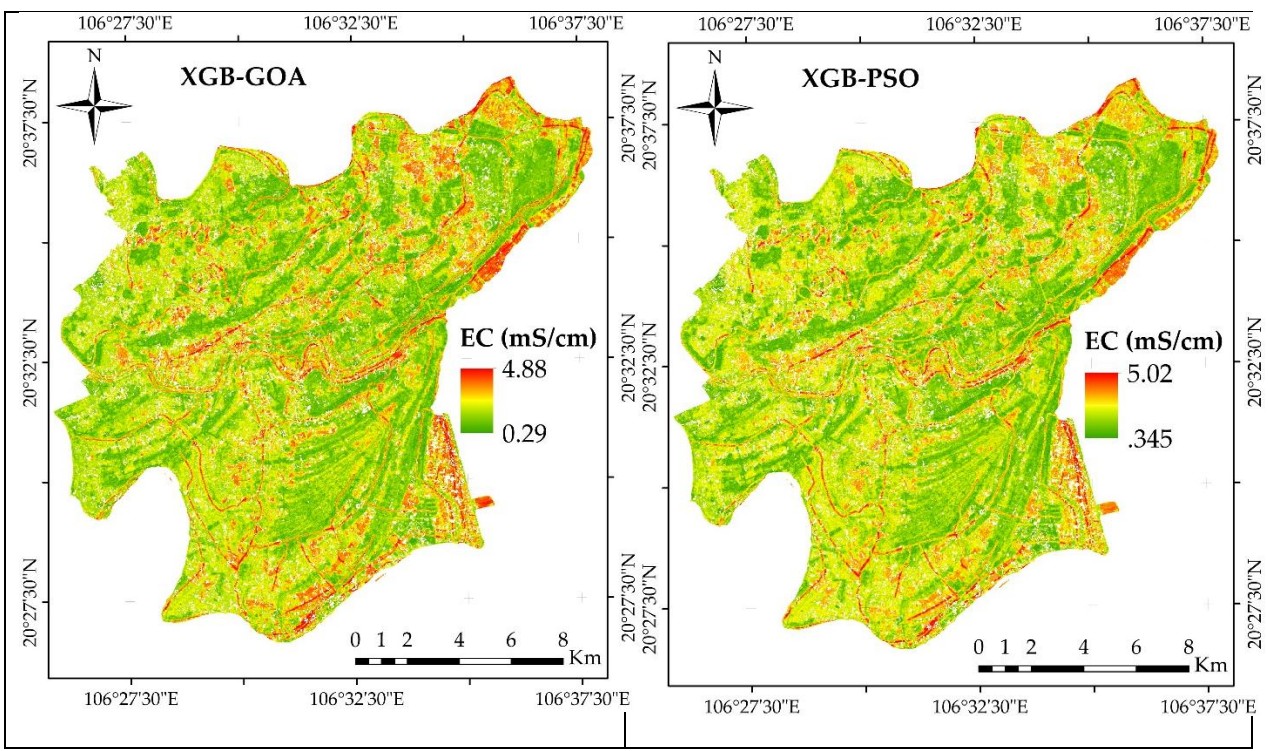

Figure 5: Soil salinity mapping in the Thai Thuy district

## 4.4. Farmers' Adaptive Capacity Assessment

Soil salinity is a key challenge for deltas worldwide, particularly in deltas where population density is high and socio-economic conditions are poor (Hoque et al., 2016). The Red River Delta is the most densely populated area in Vietnam and one of the most densely populated deltas in Southeast Asia; therefore it is crucial to consider the impact of soil salinity in this delta on the farmer's life and their adaptive capacity. Spatial distribution maps of soil salinity show that this phenomenon occurs in many areas of the Thai Thuy district, especially in coastal areas. This phenomenon certainly has a significant impact on people's living conditions and production areas, posing challenges to their livelihoods. In this section, we address the adaptive capacity of farmers in An Tan commune, a coastal area, through five elements: i) natural resources including land use, awareness of saline intrusion, perceived impacts on agricultural activities, and adaptation measures taken, ii) human resources such as household demographics, education, and farming experience, iii) physical resources, including the availability of farming equipment and infrastructure, iv) financial resources focusing on household income, income structure, credit access, and changes over time, and v) social resources addressing support from government and social networks, community cooperation, and participation in collective

adaptation activities. We interviewed 87 households in the An Tan commune to analyze the community's ability to
adapt to saline intrusion.
*4.4.1. Natural Resources*
Due to the process of salinization, we are facing a major threat to agriculture and sustaining arable land. Excess salinity
adversely influences soil structure and fertility, plant growth, crop yield, and microorganisms (Tarolli et al., 2024).
Soil salinity is frequently associated with water salinity. Groundwater in littoral regions of the Red River Delta is
characterized by high salinity (Hoque et al., 2016). The scarcity of freshwater poses significant challenges for crop
irrigation. Irrigating with saline water exacerbates soil salinity. In addition, soil salinity also creates a scarcity of
grazing land and fodder cropland of coastal areas. Coastal livestock is harshly suffering from food inaccessibility and
poultry farming in the coastal districts. All the mentioned factors impose considerable risks to the coastal inhabitant's
livelihood and food security, who rely mainly on agricultural activities such as growing rice and crops such as onions,
garlic, watermelon, and tobacco, according to our interviews.
Indeed, the results showed that 59% of interviewed households said that salinization had a medium to high impact on
agricultural production in the area in recent years, especially during the 2023 saline intrusion. Meanwhile, 38% of
households stated that saline intrusion has little or very little impact on agricultural production, mainly households in
areas far from the coast and so less affected. Households in the study area have implemented various measures to
mitigate the increasingly serious saline intrusion, such as washing the salt from the fields after each crop (as instructed
by local authorities) and adjusting their cropping systems by adopting salt-tolerant crop varieties or transitioning to
aquaculture practices. 66% of interviewed households said they had to change the crop structure to suit the saline
intrusion or switch to non-agricultural occupations to earn more income. Of the 87 households interviewed, 28% had
their main income from non-agricultural activities. A large number of people switching to non-agricultural activities
can meet their livelihood needs, but in the long-term, farmers abandoning agricultural activities to seek jobs in factories
or migrate to cities also poses many negative environmental and social consequences, such as the decline of
agrobiodiversity or labor shortages in agriculture, etc. (Subedi et al., 2022).
4.4.2. *Human Resources*
In agricultural production and the adaptability of the community to salinity intrusion, demographics are considered
one of the most important factors contributing to the creation of labour resources, directly affecting crop productivity.
The results of interviews with 87 households showed that each household has an average of 3.5 members, comprising
2.5 workers and 1 dependent member. Using available family resources reduces labor costs, thereby increasing
production profits. However, the quality of human resources poses a concern when adapting to saline intrusion. One
of the criteria for evaluation is education level. Most workers in households had a junior high school degree (66%),
5% of interviewees had a high school degree, and less than 3% had a university degree. According to previous studies,
education is an important factor in determining workers' income. In the context of climate change and sea level rise,
agriculture is negatively affected by these phenomena: low levels of education mean a lesser ability to absorb new
knowledge and methods in organizing production to reduce the negative impacts of saline intrusion. Although 83%
of the interviewed households had more than 20 years of experience in agricultural production, their knowledge of
saline intrusion and climate change was still limited. Specifically, people lack the adequate knowledge and skills to
adapt to changes in environmental conditions, leading to difficulties in choosing appropriate livelihood models.
Furthermore, saline intrusion has led to several health insecurity. Coastal residents in saline areas are at risk of
consuming high salt above the recommended levels. It is evaluated that over 7 million coastal populations in
Bangladesh, India, and Vietnam suffer from hypertension and cardiovascular diseases as a result of long-term
ingestion of saline groundwater (Hoque et al., 2016). This has profound consequences for the development and quality
of human resources in these areas.
*4.4.3. Physical Resources*
Material resources include essential items that serve people's daily life and livelihoods. The majority of the households
interviewed were engaged in agriculture, so the means of production were mainly related to agricultural activities. Of
the 87 households interviewed, 90% were equipped with agricultural production equipment such as pumps, sprayers,
and tractors, while 100% had access to tractors and harvesters for farming. The interviewed households used
equipment to exploit water sources for agricultural production; however, because the town of An Tan is located in a
coastal area, groundwater and surface water are often affected by saline intrusion. Thus, they still faced challenges
related to water resources, especially in the context of saline intrusion
Moreover, in response to saline intrusion, accessing freshwater for daily use and irrigation often leads to the
spontaneous extraction of groundwater through tubewells, a common practice in coastal areas of Vietnam and
Southeast Asia (Hoque et al., 2016). However, excessive groundwater extraction and improper irrigation practices
also pose many potential risks of increasing water resource depletion and accelerating salinization processes (Tarolli
et al., 2024). This will likely undermine the long-term adaptive capacity of coastal communities.
*4.4.4. Financial Resources*
The interviews with 87 households demonstrated that 9% of the households interviewed were poor and near-poor. It
can be seen that economic status greatly affects people's ability to adapt and recover from soil salinity. Poor and near-
poor households frequently have more difficulty evaluating solutions to mitigate the impact of soil salinity on
agricultural production. The primary source of income for a large part of the population mainly comes from
agricultural activities: 72% of the households interviewed confirmed that their main livelihood was agriculture.
However, their agricultural income is frequently unstable. This stresses the vulnerability of people's livelihoods due
to the strong effects of soil salinity on agricultural activities. Meanwhile, 56% had a stable source of income from
factory work. This emphasizes the need to diversify income sources for inhabitants in soil salinity areas. In addition,
although the income from agricultural production was enough to cover farmers' daily life, most of the households
interviewed could not save. Therefore, with increasing saline intrusion in the context of climate change, it is very
difficult for these households to have an effective response or adaptation solutions. Borrowing capital to overcome
the negative impacts of saline intrusion is one of the adaptation strategies reported by the interviewed households.
36% of households borrowed capital from relatives, 11% borrowed from credit funds or banks, 14% from local
organisations, and 33% from distribution agents. Meanwhile, 8% of the interviewed households could not borrow any
capital to overcome the consequences of saline intrusion.
Diversifying external sources of capital can help households overcome the consequences of saline intrusion and
support agricultural production more generally. However, there are still several people who cannot access capital
sources, and training to the adaptability and resilience of the people. This increases the impact of soil salinity on the
community in the study area. Furthermore, a capital utilization strategy must be carefully considered to ensure efficient
use of resources to improve livelihoods and enhance adaptability to saline intrusion.
4.4.5. *Social Resources*
Social resources play a key role in mitigating the impacts of soil salinity on the adaptability and resilience of the
people. In the study area, households received support from various sources, such as local communities, volunteer
organizations, non-governmental organizations, and mutual assistance among households. This support includes
exemption from land use tax, support for production equipment, crops, and food supply for people. Regarding people's
awareness of climate change and its impact on salinity intrusion, about 82% of households said they learned about
this issue through local authorities and media propaganda.. Furthermore, 100% of the households surveyed reported
that local authorities also had organised training sessions and drills to respond to saline intrusion and sea level rise.
However, as mentioned above, the knowledge and skills of inhabitants are still limited. This raises questions about
the effectiveness of training sessions. In addition, these training activities occur infrequently. For example, during the
soil salinity, people did not receive timely support and assistance from these organisations. This led to a reduction in
the community's ability to adapt soil salinity.
**5. Discussion**
Soil salinity is a global environmental threat, a key cause of food insecurity worldwide (Song et al., 2024). Therefore,
it is essential to monitor it with high precision, as is identifying the adaptive capacity of those in vulnerable regions.
In Vietnam, two large deltas ensure food security not only, but also in other countries. Although several previous
studies have been conducted to assess soil salinity in Vietnam, most have focused on assessing soil salinity and
farmers' adaptive capacity in the Mekong Delta (Hoang and Hai, 2024; Nguyen et al., 2024). Research on the Red
River Delta is scarce. The Red River Delta is one of the key agricultural regions in Southeast Asia. Therefore, assessing
soil salinity and farms' adaptive capacity in this area is necessary. In this study, remote sensing, machine learning,
and community interviews were used to evaluate soil salinity and the adaptive capacity of farms in the delta.
Remote sensing plays a key role in analyzing soil salinity because the salt in the soil has a significant effect on the
spectral reflectance of the soil. Soils with different salinity levels will have different spectral characteristics; for
example, areas covered with white salt often have higher spectral reflectance levels and salinity (Hoa et al., 2019; Wu
et al., 2018; Xiao et al., 2023). This is the basis for using remote sensors to monitor saltwater intrusion. However, one
of the challenges in using Sentinel 2 satellite images in soil salinity monitoring is that sometimes, the spectral
reflectance level is not consistent with soil salinity. Many studies have integrated vegetation indices in soil salinity
monitoring to minimize this limitation because different areas will have different soil salinity. It should also be noted
that the difference depends on the vegetation type in each area. Therefore, this study has integrated Sentinel 1 images
in soil salinity monitoring. Sentinel 1 images use radar signals to monitor moisture and dielectric properties providing
accurate information on soil salinity. This is particularly important in coastal areas, where surface moisture is high,
reducing the accuracy of optical imagery. This approach identifies areas severely affected by salinity intrusion while
supporting the assessment of the adaptive capacity of communities in the area (Hoa et al., 2019). However, with the
increase in the volume, type, and speed of remote sensing data collection, bottlenecks in the data analysis process may
occur (due to the inadequacy of the structure of current models for processing large datasets).
XGB is one of the most powerful algorithms for identifying the spatial distribution of natural hazards, such as floods,
landslides, and soil salinity. Its advantages include the ability to avoid the overfitting problem and fast convergence.
Additionally, XGB effectively handles missing values (Liu et al., 2022; Mo et al., 2019). However, both the
configuration and interpretation of XGB are more complex, and the parameters of this model are also complex to tune.
Incorrect parameter selection can reduce performance (Ramraj et al., 2016). Therefore, it is necessary to use
optimization algorithms to select the parameters of this model. In this study, five optimization algorithms, namely
POA, STO, SOA, PSO, and GOA, were utilized to optimize the parameters of XGB. The XGB-POA model
outperformed the other models as it is easy to carry out, has few parameters to adjust, has faster convergence capability,
and can avoid local minima - which enables it to find the best global solution (Premkumar and Santhosh, 2024).
Previous studies have indicated that POA also can solve complex problems with a large number of variables and non-
linear properties (Alamir et al., 2023; Li et al., 2023). XGB-STO model ranked second. The STO algorithm maintains
a good equilibrium with exploration and exploitation processes. This allows it to avoid local minima problems, which
improves model performance (Al-Sarray et al., 2024; Trojovský et al., 2022). The XGB-SOA model came third in
terms of accuracy. SOA can solve complex problems with a large number of variables or continuous, discrete, or
multi-objective problems, so it is a versatile tool for several different applications. In addition, inspired by the serval's
precise jumps and fast movements, SOA can converge quickly with high accuracy (Dehghani and Trojovský, 2022;
Sindi et al., 2024). The XGB-PSO model was ranked fourth. In addition to ease of use, PSO has the advantage of
equilibrium of the exploration and exploitation processes. This can avoid the local optimization problem (Juneja and
Nagar, 2016; Rini et al., 2011). The XGB-GOA model was less accurate than other models because it tends to
concentrate exploration at the beginning of the process to avoid the local optimization problem. This may lead to slow
convergence (Mirjalili et al., 2018; Zhao et al., 2019). When comparing the models proposed in this study on the
ability to predict natural hazards such as soil salinity, each model has different characteristics that influence the real-
time prediction ability. Three models (XGB-POA, XGB-STO, and XGB-SOA) can converge quickly because of the
faster learning speed. Therefore, these models best suit adaptation for real-time applications because fast updates are
necessary to support those tasked with developing mitigation strategies.
The results of this study not only confirmed the effectiveness of XGBoost models in soil salinity prediction but also
showed the potential for improving accuracy by combining them with optimization algorithms. Compared with
previous studies, the models in this study outperform traditional models. Wang and Sun (2024) used three machine
learning models, namely random forest (RF), support vector machine (SVM), and artificial neural network (ANN), to
predict the soil salinity in Huludao City, China. The results indicated that the RF model performed better with an $R^2$
value of 0.84. The model accuracy in Wang and Sun's 2024 study was lower than the models in our study. Aksoy et
al. (2024) applied two models, namely XGB and RF to predict the soil salinity in western and southeastern Lake Urmia
Playas (LUP) in the Northwest of Iran. The results showed that the XGB model was more efficient with the $R^2$ value
of 0.83. It was less accurate than in our study. Elshewy et al. (2024) evaluated the soil salinity in Sharkia Governorate,
Egypt, using four machine learning models, namely support vector machines (SVM), regression trees, Gaussian linear
regression, and tree-based ensemble. The results indicated that the SVM model performed better with an $R^2$ value of
0.86. Comparison with previous studies showed the potential of the machine learning model in this study to predict
soil salinity.
Saline intrusion in the Red River Delta and the study area reflects the interaction between natural factors and human
activities. One cause of the increasingly serious saline intrusion in the study area is the reduced flow in the delta caused
by the construction of dams and reservoirs upstream in China. This reduction reduces the ability of the river system
to repel salt water, creating conditions for seawater to penetrate deep into the inland. Hien et al. (2023) have
emphasized that by 2050, saltwater intrusion is likely to extend about 20 km inland from the river mouth, related to
sea level rise and reduced discharge from the upper river. Nguyen et al. (2017) reported that the increasing trend of
saline intrusion is the result of sea level rise, combined with the decline of the Red River water level, especially in the
dry season. Specifically, the sea level increased by 0.19 m in the period 1901-2010, with an average rate of 3.2 m from
1993-2010. In addition, the phenomenon of saline intrusion is increasingly severe due to subsidence related to
groundwater exploitation. In many areas of the Red River Delta in general and the study area in particular, uncontrolled
groundwater exploitation for agricultural production and aquaculture contributes to subsidence, increasing the impact
of tides. Nguyen and Takewaka (2020) have emphasized that the subsidence phenomenon in the delta can reach -12.3
mm/year, which is one of the causes that aggravate the problem of saltwater intrusion, especially in the context of
rising sea levels.
The results of this study explored the adaptive capacity of farms in the Thai Thuy district of Thai Binh province.
Riverine farmers in areas affected by saltwater intrusion are prepared. They rely on their local communities and expect
support from local authorities and voluntary organisations. Our results are similar to those of previous studies
investigating the adaptive capacity of residential communities to natural hazards, including saltwater intrusion. The
key to adaptation is education, knowledge, and resources to cope with saltwater intrusion. These resources can be
natural, physical, financial, social, and human resources.
The community's adaptive capacity in the study area faces many challenges, especially in the context of global
warming and growing saltwater intrusion. Although most households surveyed have more than 20 years of experience
in agricultural production and benefit from available labor resources, their adaptive capacity to saltwater intrusion
remains limited. This is in part because these households lack the knowledge to change their livelihood patterns in
addressing varying environmental situations. In addition, the main sources of agricultural income are often unstable,
and the ability to accumulate finances is low, leading to difficulties in adapting to and recovering from saltwater
intrusion. People's adaptation strategies, such as uncontrolled groundwater extraction and conversion to non-
agricultural activities, also present long-term environmental and social risks. Furthermore, policies and support
programs for residents, such as training sessions and lending programs provided by stakeholders, also raise concerns
regarding their effectiveness and inclusiveness. Although people in the study area have access to capital from many
different sources, some households still cannot access these sources of capital to overcome the consequences of
saltwater intrusion. All of these factors impact agriculture and human life, leading to increased household
vulnerability. To enhance people's adaptive capacity, it is important to emphasize the role and effectiveness of policies
of local governments, policymakers and stakeholders in supporting people to understand better and respond to saline
intrusion. Information and knowledge sharing can be done through direct outreach to people to raise awareness of
saline intrusion among communities. Lending policies of local governments and stakeholders need to cover all
households while improving the efficiency of capital use. Effective management of natural and physical resources and
enhancing social capital through the development of cooperative community models are important factors contributing
to people's adaptive capacity to saline intrusion. This study has successfully built a theoretical framework using
machine learning with optimization algorithms, remote sensing, and farmer interviews to determine the spatial
distribution of soil salinity and farmers' adaptation capacity. However, to apply this theoretical framework in different
regions, it is necessary to use factors specifically pertinent to each region. Machine learning models must be provided
with the local characteristics of the region in question. However, data collection in any region is difficult, often due to
restrictive data-sharing policies or limited financing resources to maintain and distribute the data.
From field surveys, it can be seen that in the Red River Delta, adaptation options to soil salinity mainly rely on
upgrading the sea dike system, river dikes, and saline prevention sluice systems. In addition, other adaptation options
mainly include increasing the resilience of the current agricultural system, such as changing the crop calendar,
changing crop varieties, using fertilizers, and planting mangroves. Many households have transitioned from rice
cultivation to aquaculture in coastal areas, where soil salinity has a significant impact. In addition, some fish farming
households have also switched to shrimp farming or fish farming due to increased saline intrusion. Some households
do not have the capital to convert their agricultural systems, and while agricultural productivity decreases due to saline
intrusion, they consider finding non-agricultural jobs or migrating to the city to find jobs with more stable incomes.
Households located further inland, less affected by saline intrusion, still maintain traditional agriculture. Some
households practice intercropping by growing rice alongside vegetables to increase their income. Thus, it can be seen
that the adaptability of households in the Red River Delta is not only based on strengthening the system of sea dikes,
river dikes, and salinity prevention sluices, but also on transforming the traditional agricultural system to minimize
the impact of salinity intrusion. However, capital barriers force many households to abandon agriculture, seriously
affecting the food security situation in the region.
A significant problem when using machine learning is that of extrapolation. Each model built is adapted only to one
set of data. Therefore, evaluating the soil salinity in other regions is challenging. General, there is only one model
that fits each training dataset. In theory, this would not be a problem if enough training data were collected and all
extreme events were included. However, in practice, it is very difficult to collect data for all these events, especially
in the context of climate change and sea level rise.. To solve this problem, several studies have pointed out that
integrating machine learning with conventional models for example, remote sensing or hydrodynamic models can be
effective, as such traditional models can provide the training data to use as the input file of the machine learning
model. Another solution is to combine machine learning with optimization algorithms, as in this study, to enhance the
prediction capability of the machine learning model (Tran and Kim, 2022).
This study emphasizes the significance of combining machine learning methods to analyze the spatial distribution of
salinity intrusion with the community's adaptive capacity to soil salinity. The salinity intrusion map from the machine
learning model shows a clear difference in the level of salinity intrusion between coastal, riverside, and inland areas.s.
Coastal and estuarine areas often have high levels of salinity intrusion, with EC values exceeding 7 mS/cm. These are
also areas where communities must apply appropriate adaptation strategies, including crop restructuring, selecting
more salinity-tolerant plant varieties, combining agriculture and fisheries, or livelihood conversion. In contrast, inland
areas, where the level of salinity intrusion is lower, have less variation in agricultural production models, and
communities in these areas still mainly maintain traditional agricultural practices. The findings may indicate that the
coping strategies and adaptive capacity of the communities depend on the level of salinity intrusion in the areas. In
addition, it can be seen that in areas with high salinity intrusion, people have difficulty in accessing fresh water for
agricultural production; therefore, the communities in this area tend to depend more on non-agricultural sources of
income. Previous studies (Nguyen et al., 2019; Yuen et al., 2021) have demonstrated this trend.
The results of the study emphasize that the integration of advanced machine learning models and sociological surveys
not only improves the comprehensive research ability from natural factors to socio-economic factors but can also
support policymakers and planners to develop appropriate adaptation solutions. Identifying areas affected by saline
intrusion by using machine learning models and qualitative analysis of the adaptive capacity of the community is a
solid scientific basis for developing policies to minimize the impact of saline intrusion, especially in the context of
climate change, to ensure agricultural development and food security.
This study was successful in building machine learning models integrated with optimization algorithms to identify the
spatial of soil salinity, as well as evaluating farmers' adaptive capacity in the study area. However, in terms of data,
this study collected 62 soil salinity samples to build the machine learning model; therefore, the soil salinity map
constructed by the proposed models cannot present the trend and drive of soil salinity in time series. Furthermore, soil
salinity is significantly affected by climate change and rising sea levels, so it is necessary to assess the effects of this
change on soil salinity in the future.
As hydrological conditions change, those living in deltas are confronting increased risk. The Red River Delta is one
of the largest deltas in the world and, thanks to its fertile floodplains, is home to about 21 million inhabitants. In recent
years, in the context of global warming and rising sea levels, these deltas are confronting growing flooding and soil
salinity problems, which affect food security in the region and the country. Policies must be implemented to improve
the agricultural system and the adaptive capacity of farmers. A proactive approach is required, envisaging multiple
scenarios to provide appropriate support for agriculture. These scenarios may include activities and programs adaptive
to the different influences of global warming on soil salinity.
**6. Conclusion**
Soil salinity is a key environmental threat, which will have a growing effect on the development of agriculture and
food security globally. A lack of assessment of local adaptive capacities exacerbates the problem. Therefore, this
research's objective was toconstruct a theoretical framework to assess soil salinity and farmers' adaptive capacity
based on machine learning, optimization algorithms, remote sensing, and interviews with local people. The results in
this study represent a novel contribution to the literature for researchers worldwide and can support policy-makers and
farmers to establish suitable strategies to limit damage related to soil salinity. The outcome of this research is as
follows.
-  This study justified the effectiveness of machine learning and remote sensing in soil salinity monitoring in the Red
River Delta. The results of this study can be opened to realize in different regions.
- Five optimization algorithms, namely POA, STO, SOA, PSO, and GOA, were successful in optimizing the accuracy
of the XGB model. All these algorithms were successful in improving the accuracy of XGB. Of these, the XGB-POA
model showed the greatest performance, with an $R^2$ value of 0.968. This was followed by XGB-STO ($R^2$=0.967),
XGB-SOA ($R^2$=0.966), XGB-PSO ($R^2$=0.964), and XGB-GOA ($R^2$=0.964).
- The models in this research were utilized to construct soil salinity maps. The maps demonstrated that littoral areas
and those along the rivers were the most influenced by the soil salinity problem because these regions are influenced
by seawater. In addition, when the river levels are lower during the dry season, it creates the conditions for seawater
to penetrate the land.
- Five factors were analyzed to consider farmers' adaptive capacity: natural capital, human capital, material resources,
financial resources, and social capital. The results show that people have awareness and actions in improving their
adaptive capacity to increasingly severe saline intrusion; however, there are still many limitations related to lack of
awareness and finance. As a recommendation, the participation of multiple stakeholders is required, with a particular
emphasis on the role of policies in sustainably and effectively enhancing people's adaptive capacity.
The outcome of this research provides key knowledge on the spatial distribution of soil salinity and farmers' adaptive
capacity to growing salinization, to support local authorities or farmers in proposing appropriate measures to reduce
soil salinity damage. This can complement a theoretical framework in the existing literature on soil salinity
management and adaptive capacity.
**Statements and Declarations**
**Funding**
This research is funded by Vietnam National Foundation for Science and Technology Development (NAFOSTED)
under grant number 105.08-2023.13
**Competing Interests**
The authors declare that they have no conflict of interest.
**Ethics and consent to participate**
Not applicable
**Consent for publication**
Not applicable
**Authors Contributions**
**Huu Duy Nguyen:** Conceptualization, Formal analysis, Funding acquisition, Investigation, Methodology,
Supervision, Writing – original draft, Writing – review & editing. **Quang-Thanh Bui:** Conceptualization,
Formal analysis, Investigation, Methodology, Supervision, Writing – original draft, Writing – review &
editing. **Thi Anh Tam Lai:** Data curation**. Duc Dung Tran:** Data curation, Investigation, Writing – original
draft, Writing – review & editing. **Dinh Kha Dang:** Data curation, Investigation. **Himan Shahabi:** Writing
– original draft, Writing – review & editing.
**Availability of data and materials**
The datasets used and/or analysed during the current study available from the corresponding author on reasonable
request

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
