# Peer review of "Predicting Soil Salinity in the Red River Delta (Vietnam) Using Machine Learning and Assessing 2 Farmers' Adaptive Capacity Huu Duy Nguyen1, Dinh Kha Dang2, Thi Anh Tam Lai1, Duc Dung Tran3, Himan Shahabi4, Quang-3 4 Thanh Bui1"

_EGUsphere, 2025_

## Referee Comment (RC2)

Review paper: **Farmers' adaptive capacity towards soil salinity effects using hybrid machine learning in the Red River Delta**

Code: **egusphere-2025-1051**

**General comments:**

Soil salinity, which significantly impacts agricultural activities worldwide, is considered one of the major environmental hazards caused by both natural and human-induced processes. This phenomenon has become increasingly severe due to the impacts of climate change, particularly rising sea levels. Therefore, evaluating soil salinity is regarded as a critical task for supporting sustainable agricultural planning. Additionally, evaluating adaptive capacity is considered an essential tool to mitigate the effects of soil salinity on community livelihoods. One of the strengths of this article is the integration of physical data, machine learning models, and socio-economic data (through interviews with local populations). As such, this article is highly relevant and well-aligned with the journal's scope. I accept to publish this article with the condition of major revisions.

**Main comments:**

Abstract: Although the authors present the objectives, data, and results of the article, I would like to see the inclusion of quantitative results and the significance of the findings.

Introduction: It is necessary to highlight the significance of this article. Additionally, it is important to emphasize the role of adaptive capacity in reducing the effects of soil salinity.

Study Area: The reasons for selecting this study area should be explained in more detail, especially the effects of soil salinity on agricultural activities.

Map 1: Please revise Map 1 for better clarity.

Map 2: Similarly, Map 2 should be revised for better quality.

Methodology: This study uses machine learning and optimization algorithms to construct the soil salinity map. However, I do not fully understand how the authors constructed these models. A more detailed explanation is needed.

Interviews with Local Populations: The inclusion of the interview methodology is necessary because adaptive capacity is a key outcome.

Discussion: Although this article clearly discusses the strengths and weaknesses of the machine learning models, and also touches on the adaptive capacity of the populations, I believe it would be useful to add the methodology for addressing the effects of soil salinity at the community level.

Extrapolation Issues: In this section, the authors present issues of extrapolation. I would suggest expanding on this point, as it is a challenge not only in soil salinity but also in other types of natural hazards.

---

## Author Comment (AC1)

**Response to reviewer's comments**

We'd like to thank the editor for giving us a chance to revise the manuscript. We also would like to thank the reviewers for spending their expensive time and expertise to comment on our manuscript. We have carefully read the comments and revised the manuscript accordingly, point by point. Based on these comments, we think that the quality of the manuscript has been very much improved and now meets the journal standard. All the changed texts in the revised manuscript are marked with **red color** in a separate file for reviewing.

**Reviewer 2**

Soil salinity, which significantly impacts agricultural activities worldwide, is considered one of the major environmental hazards caused by both natural and human-induced processes. This phenomenon has become increasingly severe due to the impacts of climate change, particularly rising sea levels. Therefore, evaluating soil salinity is regarded as a critical task for supporting sustainable agricultural planning. Assessing adaptive capacity is also regarded as a crucial instrument for reducing the impact of soil salinity on local livelihoods. One of the strengths of this article is the integration of physical data, machine learning models, and socio-economic data (through interviews with local populations). As such, this article is relevant and well-aligned with the journal's scope. I accept publishing this article with the condition of major revisions.

Response: We would like to thank the reviewer for the constructive and valuable comments. We have revised the manuscript's content based on your feedback and addressed them all. The manuscript's quality has been significantly improved after the revision. Going forward, our responses in blue are to try our best to ensure clarity and efficiency, but please know that we genuinely appreciate all the reviewer's inputs.

Abstract: Although the authors present the objectives, data, and results of the article, I would like to see the inclusion of quantitative results and the significance of the findings.

Response : Thank you for your observation. We have added the quantitative resultat in the abstract.

Introduction: It is necessary to point out the importance of this article. Additionally, it is important to emphasize the role of adaptive capacity in reducing the effects of soil salinity.

Response : Thank you for your observation. We have added the information due to the role of adaptive capacity in line 116-127.

"The adaptive capacity is defined as the capability of the community to cope, adjust, and adapt to the impacts of growing soil salinity. It measures the ability to predict, respond, and recover from the phenomenon. It is assessed on different scales, using different approaches, according to the region in question (Mazumder and Kabir, 2022; Thiam et al., 2024). Furthermore, understanding the adaptive capacity of communities plays an important role in reducing the negative effects of salinity intrusion in coastal regions in general and the Red River Delta in particular. By assessing adaptation at multiple scales with site-specific methods, researchers and local governments can identify interventions (such as crop variety changes, crop calendars, irrigation systems) that are effective. The IPCC in 2014 indicated that farm adaptive capacity depends on five main factors: natural capital, human capital, material resources, financial resources, and social capital. Therefore, integrating the adaptive capacity of populations with the soil salinity map

improves the accuracy of predictions and proposes adaptation strategies that strengthen the overall resilience of communities."

Study Area: The reasons for selecting this study area should be explained in more detail, especially the effects of soil salinity on agricultural activities.

Response : Thank you for your observation. We have added the information due to the effects of soil salinity on agricultural activities as suggested in line 180-186

Map 1: Please revise Map 1 for better clarity.

Response: Thank, corrected.

Map 2: Similarly, Map 2 should be revised for better quality.

Response: Thank, corrected.

Methodology: This study uses machine learning and optimization algorithms to construct the soil salinity map. However, I do not fully understand how the authors constructed these models. A more detailed explanation is needed.

Response: Thank you for your observation. We have added the information due to explaine how they are integrated with the XGBoost model with the algorithm optimisation in line 270-291.

« The machine learning model-building process was divided into two main steps: the first was the XGB model building, and the second was the hybrid model building (the integration of XGB with optimization algorithms). The accuracy of the machine learning model depends on the parameter adjustments of the XGB model. In this study, the XGB model parameters were selected using the trial-and-error method. Finally, the XGB parameters were n_estimators=100, max_depth=4, subsample=0.5, and colsample_bytree=0.5. While the hybrid model was built by integrating the XGB model and optimization algorithms, namely GOA, POA, SOA, STO, and PSO. To integrate the XGB model with optimization algorithms, we first need to construct an objective function F(θ) that returns the error value of XGB on the validation set when using the parameter sets θ. That is, each parameter set has a different error value. Next, determine the search space of the hyperparameters (n_estimators, max_depth, subsample, colsample_bytree) as discrete value intervals. Then, the optimization algorithms will initialize the population of individuals with the size and parameters characteristic of each optimization algorithm. This study was tested with 500 iterations: at each iteration, each individual will generate a combination of θi, and the optimization algorithms will update the position or velocity of the individuals according to their own rules. This process is repeated until a stopping threshold is set. Finally, the results are the optimal parameters. The parameters of the model are as follows: problem_size = 3,   batch_size = 25, epoch = 500, pop_size = 50, "fit_func": fun_avr2, "lb": [0] problem_size, "ub":

[1] problem_size, c_min = 0.00004, c_max = 2.0 for XGB-GOA; problem_size = 3, batch_size = 25, epoch = 500, pop_size = 50, "fit_func": fun_avr2,   "lb": [0] problem_size, "ub": [1] problem_size, c1=2.05, c2=2.05, w_min=0.4 for XGB-PSO ; problem_size = 3, batch_size = 25, epoch = 500, pop_size = 50, "fit_func": fun_avr2, "lb": [0] problem_size, "ub": [1] problem_size  for XGB-POA; problem_size = 3, batch_size = 25, epoch = 500, pop_size = 50, "fit_func": fun_avr2, "lb": [0] problem_size, "ub": [1] problem_size for XGB-SOA; problem_size = 3, batch_size = 25, epoch = 500, pop_size = 50, "fit_func": fun_avr2, "lb": [0] problem_size, "ub": [1] problem_size for XGB-STO.»

Interviews with Local Populations: The inclusion of the interview methodology is necessary because adaptive capacity is a key outcome.

Response: Thank you for your observation. We have corrected the interview methodology.

Discussion: Although this article clearly discusses the strengths and weaknesses of the machine learning models and also touches on the adaptive capacity of the populations, I believe it would be useful to add the methodology for addressing the effects of soil salinity at the community level.

Response: Thank you for your observation. We have added the information due to the strategie to reduce the effects of soil salinity in line 704-717.

"From field surveys, it can be seen that in the Red River Delta, adaptation options to soil salinity mainly rely on upgrading the sea dike system, river dikes, and saline prevention sluice systems. In addition, other adaptation options mainly include increasing the resilience of the current agricultural system, such as changing the crop calendar, changing crop varieties, using fertilizers, and planting mangroves. Many households have transitioned from rice cultivation to aquaculture in coastal areas, where soil salinity has a significant impact. In addition, some fish farming households have also switched to shrimp farming or fish farming due to increased saline intrusion. Some households do not have the capital to convert their agricultural systems, and while agricultural productivity decreases due to saline intrusion, they consider finding non-agricultural jobs or migrating to the city to find jobs with more stable incomes. Households located further inland, less affected by saline intrusion, still maintain traditional agriculture. Some households practice intercropping by growing rice alongside vegetables to increase their income. Thus, it can be seen that the adaptability of households in the Red River Delta is not only based on strengthening the system of sea dikes, river dikes, and salinity prevention sluices, but also on transforming the traditional agricultural system to minimize the impact of salinity intrusion. However, capital barriers force many households to abandon agriculture, seriously affecting the food security situation in the region."

Extrapolation Issues: In this section, the authors present issues of extrapolation. I would suggest expanding on this point, as it is a challenge not only in soil salinity but also in other types of natural hazards.

Response: Thank you for your observation. We have added the discussion due to the extrapolation issues in line 718-727.

"A significant problem when using machine learning is that of extrapolation. Each model built is adapted only to one set of data. Therefore, evaluating the soil salinity in other regions is challenging. General, there is only one model that fits each training dataset. In theory, this would not be a problem if enough training data were collected and all extreme events were included. However, in practice, it is very difficult to collect data for all these events, especially in the context of climate change and sea level rise.. To solve this problem, several studies have pointed out that integrating machine learning with conventional models for example, remote sensing or hydrodynamic models can be effective, as such traditional models can provide the training data to use as the input file of the machine learning model. Another solution is to combine machine learning with optimization algorithms, as in this study, to enhance the prediction capability of the machine learning model (Tran and Kim, 2022)."

---

## Author Comment (AC2)

**Response to reviewer's comments**

We'd like to thank the editor for giving us a chance to revise the manuscript. We also would like to thank the reviewers for spending their expensive time and expertise to comment on our manuscript. We have carefully read the comments and revised the manuscript accordingly, point by point. Based on these comments, we think that the quality of the manuscript has been very much improved and now meets the journal standard. All the changed texts in the revised manuscript are marked with **red color** in a separate file for reviewing.

**Reviewer: 1**

1. Title

The current title may not accurately reflect the study's output. In the present status, the study does not use machine learning to assess farmer's adaptive capacity, but rather to predict soil salinization. The title should be reconsidered and rephrased to avoid any misleading interpretations.

**Responses:**

Thank you for your observation. We have changed the title of the manuscript with the new title "Predicting Soil Salinity Using Machine Learning and Assessing Farmers' Adaptive Capacity: A Study in the Red River Delta".

2. Astract

The abstract is complete and gives a clear idea of the content without reading the paper.

**Response:**
Thank you for your attention.

3. Introduction

Overall, the introduction covers the state of the art and explains the objectives of the study in a complete way. However, several acronyms and abbreviations are introduced here without first presenting their full forms. I recommend carefully reviewing the Introduction, and the manuscript as a whole, for consistency in defining all acronyms upon first use.

**Response:**
Thank you for your valuable comment. We have corrected entire the introduction section. Please refer to the revised manuscript.
Minor comments:

L42: Use "posing" instead of "poses".

**Response:**

Thank you for your attention. We have corrected it in line 45.

L51: Please rephrase "represent an extremely key role".

**Response:**

Thank you for your comment. We have corrected it in line 55-62. We copied here for fast reviewing:

"This problem is increasingly serious in Mekong Delta and Red River Delta - home to over 40 million people and playing a key role in Vietnam's agricultural and aquaculture sectors - where they account for 71% of paddy cultivation, 86% of aquatic farming, and 65% of fruit production (General Statistics Office, 2024; Ministry of Aquaculture, Agriculture and Rural Development, 2013). Because these low-lying coastal areas (Hung and Larson, 2014) are experiencing subsidence (Le Dang et al., 2014), and declining river water levels,, they have become highly susceptible to the effects of climate variability and sea-level rise (Dasgupta et al., 2009)."

L126: This passage would be more suitable for the final remarks (Conclusion) that the Introduction.

**Response:**

Thank you for your observation. We have deleted this passage and we have added the novel passage to be more suitable with the introduction and highlighted the significance of the manuscript. We copied here for fast reviewing (line 148-157):

"In general, salinity intrusion harms agricultural development and people's livelihoods. Therefore, it is necessary to develop a theoretical framework to address the soil salinity problem in terms of natural and social factors. However, previous studies have mainly assessed the spatial distribution of salinity or the community's adaptive capacity, and hardly any studies have assessed both the spatial distribution of salinity and the community's adaptive capacity. Thus, the global contribution of this study is to fill the knowledge gap about the spatial distribution of soil salinity and the adaptive capacity of communities in the Red River Delta in general and Thai Binh Province in particular by relying on modern methods to improve this important and understudied understanding. The results of this study can play an important role in mitigating the impact of salinity intrusion on agricultural development and can help policymakers and planners develop effective strategies to mitigate this impact, especially in the context of climate change."

Materials and methods

The section is clearly structured into different sub-sections and easy to follow. However, some key information is unclear or missing:

- Model selection and integration: The rationale behind the selection of the specific optimization algorithms (POA, STO, SOA, PSO, GOA) and how they are integrated with the XGBoost model is not clearly explained. It is also not fully clear how these hybrid models contribute to the generation of soil salinity maps. Clarifying this connection would strengthen the methodological transparency.

**Response:**

Thank you for your valuable comment. We have added the information due to explain how they are integrated with the XGBoost model with the algorithm optimisation in line 255-276. We copied here for fast reviewing (line 270-291):

« The machine learning model-building process was divided into two main steps: the first was the XGB model building, and the second was the hybrid model building (the integration of XGB with optimization algorithms). The accuracy of the machine learning model depends on the parameter adjustments of the XGB model. In this study, the XGB model parameters were selected using the trial-and-error method. Finally, the XGB parameters were n_estimators=100, max_depth=4, subsample=0.5, and colsample_bytree=0.5. While the hybrid model was built by integrating the XGB model and optimization algorithms, namely GOA, POA, SOA, STO, and PSO. To integrate the XGB model with optimization algorithms, we first need to construct an objective function $F(\theta)$ that returns the error value of XGB on the validation set when using the parameter sets $\theta$. That is, each parameter set has a different error value. Next, determine the search space of the hyperparameters (n_estimators, max_depth, subsample, colsample_bytree) as discrete value intervals. Then, the optimization algorithms will initialize the population of individuals with the size and parameters characteristic of each optimization algorithm. This study was tested with 500 iterations: at each iteration, each individual will generate a combination of $\theta i$, and the optimization algorithms will update the position or velocity of the individuals according to their own rules. This process is repeated until a stopping threshold is set. Finally, the results are the optimal parameters. The parameters of the model are as follows: problem_size = 3,    batch_size = 25, epoch = 500, pop_size = 50, "fit_func": fun_avr2, "lb": [0] *problem_size, "ub": [1]* problem_size, c_min = 0.00004, c_max = 2.0 for **XGB-GOA**; problem_size = 3, batch_size = 25, epoch = 500, pop_size = 50, "fit_func": fun_avr2,    "lb": [0] *problem_size, "ub": [1]* problem_size, c1=2.05, c2=2.05, w_min=0.4 for **XGB-PSO** ; problem_size = 3, batch_size = 25, epoch = 500, pop_size = 50, "fit_func": fun_avr2, "lb": [0] *problem_size, "ub": [1]* problem_size  for **XGB-POA**; problem_size = 3, batch_size = 25, epoch = 500, pop_size = 50, "fit_func": fun_avr2, "lb": [0] *problem_size, "ub": [1]* problem_size for **XGB-SOA**; problem_size = 3, batch_size = 25, epoch = 500, pop_size = 50, "fit_func": fun_avr2, "lb": [0] *problem_size, "ub": [1]* problem_size for **XGB-STO**.»

- Land use consideration: It seems that the modeling process and salinity mapping does not account for different land use types. Applying models across the entire region without filtering by land use could lead to inaccurate interpretations, especially in heterogeneous agricultural landscapes. This should be explicitly addressed.

**Response:**
Thank you for your observation. We have added the information due to the land use types in line 465-483. We copied here for fast reviewing:

« According to the FAO, salinity can be divided into 5 levels: non-saline, slightly saline, moderately saline, heavily saline, and very heavily saline. In which the EC value is below 2 mS/cm, the soil is considered not saline, and the plants grow completely normally. The EC value ranges from 2- 4 mS/cm; the soil is slightly saline and has very little effect on the plants. Specifically, flower crops may grow slowly, while rice and fruit trees may have reduced height. The EC value ranges from 4-8 mS/cm; the soil is moderately saline and reduces crop yields. Specifically, rice can have a 10-20% reduction in yield. The EC value ranges from 8 - 16 mS/cm; the soil is heavily saline and affects the growth of plants. The soil becomes extremely saline and uncultivated if the EC value surpasses the threshold of 16

mS/cm. This study utilizes the XGB-POA model, which boasts the highest accuracy, to analyze the areas affected by saline intrusion. Specifically, in the study area, about 65 km² of land area is not affected by saline intrusion, 165 km² is slightly affected, and 1.8 km² is greatly affected by saline intrusion. Compared to the land cover/land use map, it can be seen that the land area greatly affected by saline intrusion is mainly aquaculture; therefore, this area will not be significantly affected in terms of productivity. Meanwhile, 165 km2 of land affected by saline intrusion is rice land, which can slow down rice growth and reduce productivity. »

 - Irrigation practices: The manuscript would benefit from including contextual information on irrigation practices in the study area. Specifically, details on the main sources of irrigation, and the distribution of land use types (e.g., paddy fields, rainfed, and irrigated areas) would provide valuable background for understanding the drivers of soil salinity and its spatial variability.

**Response:**
Thank you for your observation. We have added the information due to the irrigation practices in line 478-483. We copied here for fast reviewing:

"In the study area of Thai Thuy District, Thai Binh Province, about 70-75% of the agricultural land is irrigated by a canal system that draws water from the Red River and the Day River. Thanks to the water source from the Red River and the Day River, agricultural production areas are regularly washed away with salt, so the EC value in these areas is often below 4 mS/cm. In coastal areas, salinity intrusion at river mouths forces people to use shallow groundwater for irrigation, which leads to salt accumulation and prompts a shift from rice cultivation to aquaculture."

-  Farmer interviews: It is recommended that the authors include the full list or at least a representative sample of the questionnaire items used in the household interviews, either within the main text or as supplementary material. Moreover, the socio-economic component of the study is presented independently from the machine learning analysis, with little discussion of how the two are connected.

Response: Thank you very much for your observation. We added the information in line 496-504.

Figure 2 suggests that the selection of interview locations may have been informed by the salinity maps generated through the machine learning models, but this relationship is not clearly explained. Clarifying this linkage would enhance the coherence of the study and highlight the value of integrating spatial and social data.

Response: Thank you very much for your observation. We have added this information in lines 728-746.

"This study emphasizes the significance of combining machine learning methods to analyze the spatial distribution of salinity intrusion with the community's adaptive capacity to soil salinity. The salinity intrusion map from the machine learning model shows a clear difference in the level of salinity intrusion between coastal, riverside, and inland areas.s. Coastal and estuarine areas often have high levels of salinity intrusion, with EC values exceeding 7 mS/cm. These are also areas where communities must apply appropriate adaptation strategies, including crop restructuring, selecting more salinity-tolerant plant varieties, combining agriculture and fisheries, or livelihood conversion. In contrast, inland areas, where the level of salinity intrusion is lower, have less variation in agricultural

production models, and communities in these areas still mainly maintain traditional agricultural practices. The findings may indicate that the coping strategies and adaptive capacity of the communities depend on the level of salinity intrusion in the areas. In addition, it can be seen that in areas with high salinity intrusion, people have difficulty in accessing fresh water for agricultural production; therefore, the communities in this area tend to depend more on non-agricultural sources of income. Previous studies (Nguyen et al., 2019; Yuen et al., 2021) have demonstrated this trend.

The results of the study emphasize that the integration of advanced machine learning models and sociological surveys not only improves the comprehensive research ability from natural factors to socio-economic factors but can also support policymakers and planners to develop appropriate adaptation solutions. Identifying areas affected by saline intrusion by using machine learning models and qualitative analysis of the adaptive capacity of the community is a solid scientific basis for developing policies to minimize the impact of saline intrusion, especially in the context of climate change, to ensure agricultural development and food security."

Minor comments:

-L133: Please remove "with the".

Response: Removed

- L144: Replace "obtained at" with "reach".

Response: corrected.

-L165.166: Where are the soil sapling points located exactly?

Response: Thank you for your observation. We havec added the soil sampling points located in the figure 1

-L185: Please translate "extractés à partir de l'image" into English.

Response: Corrected

-L221: Please define what a Tan commune is.

Response: An Tan is the name of commune in the Thai Thuy District. We have corrected.

-L223: There is an extra comma "is, often".

Response: Corrected.

-L306 and onwards: Proposed by proposed by Kennedy and Eberhart (1995). Please check the reference style of similar citations throughout the manuscript.

Response: Corrected.

4. Results

-The results are clear and concise. As stated above, there is poor integration between the machine learning analysis and the socio-economic analysis. Minor comments:

-L395: What questions are asked in the interviews? (see comment above)

Response: Thank you very much. We added the information in line 496-504.

"This phenomenon certainly has a significant impact on people's living conditions and production areas, posing challenges to their livelihoods. In this section, we address the adaptive capacity of farmers in An Tan commune, a coastal area, through five elements: i) natural resources including land use, awareness of saline intrusion, perceived impacts on agricultural activities, and adaptation measures taken, ii) human resources such as household demographics, education, and farming experience, iii) physical resources, including the availability of farming equipment and infrastructure, iv) financial resources focusing on household income, income structure, credit access, and changes over time, and v) social resources addressing support from government and social networks, community cooperation, and participation in collective adaptation activities. We interviewed 87 households in the An Tan commune to analyze the community's ability to adapt to saline intrusion."

-L401: The passage "changing the crop structure" is unclear. Please rephrase.

Response: Corrected.

-L472: There is a typo here "the 2soil salinity".

Response: Corrected.

5. Discussion

-The Discussion section addresses the main findings of the study, particularly the performance of the hybrid XGBoost models and the socio-economic insights from the farmer interviews. However, it falls short in a few critical areas that limit the depth and broader relevance of the study's conclusions:

  - Lack of comparative analysis: The discussion would benefit from a more comprehensive comparison with similar studies that have applied enhanced or hybrid XGBoost algorithms (or other machine learning approaches) in soil salinity mapping or related environmental modeling tasks. Including such references would help position the study within the existing body of literature and strengthen its contribution.

Response: Thank you for your observation. We have added the discussion du to the comparative analysis in the line 646-658.

«The results of this study not only confirmed the effectiveness of XGBoost models in soil salinity prediction but also showed the potential for improving accuracy by combining them with optimization algorithms. Compared with previous studies, the models in this study outperform traditional models. (Wang and Sun, 2024) used three machine learning models, namely random forest (RF), support vector machine (SVM), and artificial neural network (ANN), to predict the soil salinity in Huludao City, China. The results indicated that the RF model performed better with an $R^2$ value of 0.84. The model accuracy in Wang and Sun's 2024 study was lower than the models in our study. (Aksoy et al., 2024) applied two models, namely XGB and RF to predict the soil salinity in western and southeastern Lake Urmia Playas (LUP) in the Northwest of Iran. The results showed that the XGB model was more efficient with the $R^2$ value of 0.83. It was less accurate than in our study. (Elshewy et al., 2024) evaluated the soil salinity in Sharkia Governorate, Egypt, using four machine learning models, namely support vector machines (SVM), regression trees, Gaussian linear regression, and tree-based ensemble. The results indicated that the SVM model performed better with an $R^2$ value of 0.86. Comparison with previous studies showed the potential of the machine learning model in this study to predict soil salinity."

- Limited interpretation of spatial variability: While the results highlight the spatial distribution of soil salinity, the discussion does not fully explore the potential environmental, agronomic, or anthropogenic drivers behind the observed variability. Possible contributing factors other than proximity to the coast or rivers should be discussed in more detail to provide context for the spatial patterns.

Response: Thank you for your observation. We have added the information due to the interpretation of spatial variability in line 659-672.

"Saline intrusion in the Red River Delta and the study area reflects the interaction between natural factors and human activities. One cause of the increasingly serious saline intrusion in the study area is the reduced flow in the delta caused by the construction of dams and reservoirs upstream in China. This reduction reduces the ability of the river system to repel salt water, creating conditions for seawater to penetrate deep into the inland. (Hien et al., 2023) have emphasized that by 2050, saltwater intrusion is likely to extend about 20 km inland from the river mouth, related to sea level rise and reduced discharge from the upper river. (Nguyen et al., 2017) reported that the increasing trend of saline intrusion is the result of sea level rise, combined with the decline of the Red River water level, especially in the dry season. Specifically, the sea level increased by 0.19 m in the period 1901-2010, with an average rate of 3.2 m from 1993-2010. In addition, the phenomenon of saline intrusion is increasingly severe due to subsidence related to groundwater exploitation. In many areas of the Red River Delta in general and the study area in particular, uncontrolled groundwater exploitation for agricultural production and aquaculture contributes to subsidence, increasing the impact of tides. (Nguyen and Takewaka, 2020) have emphasized that the subsidence phenomenon in the delta can reach -12.3 mm/year, which is one of the causes that aggravate the problem of saltwater intrusion, especially in the context of rising sea levels."

- Integration between technical and social findings: The Discussion currently treats the machine learning results and socio-economic findings as separate components. A more integrated discussion that connects spatial variability in salinity with local adaptive capacity (e.g., explaining how different levels of salinization impact farmers' strategies or vulnerability) would enhance the coherence and practical relevance of the study.

Response: Thank you very much for your observation. We have added this informations in line 699-711.

"This study emphasizes the significance of combining machine learning methods to analyze the spatial distribution of salinity intrusion with the community's adaptive capacity to soil salinity. The salinity intrusion map from the machine learning model shows a clear difference in the level of salinity intrusion between coastal, riverside, and inland areas.s. Coastal and estuarine areas often have high levels of salinity intrusion, with EC values exceeding 7 mS/cm. These are also areas where communities must apply appropriate adaptation strategies, including crop restructuring, selecting more salinity-tolerant plant varieties, combining agriculture and fisheries, or livelihood conversion. In contrast, inland areas, where the level of salinity intrusion is lower, have less variation in agricultural production models, and communities in these areas still mainly maintain traditional agricultural practices. The findings may indicate that the coping strategies and adaptive capacity of the communities depend on the level of salinity intrusion in the areas. In addition, it can be seen that in areas with high salinity intrusion, people have

difficulty in accessing fresh water for agricultural production; therefore, the communities in this area tend to depend more on non-agricultural sources of income. Previous studies (Nguyen et al., 2019; Yuen et al., 2021) have demonstrated this trend.

The results of the study emphasize that the integration of advanced machine learning models and sociological surveys not only improves the comprehensive research ability from natural factors to socio-economic factors but can also support policymakers and planners to develop appropriate adaptation solutions. Identifying areas affected by saline intrusion by using machine learning models and qualitative analysis of the adaptive capacity of the community is a solid scientific basis for developing policies to minimize the impact of saline intrusion, especially in the context of climate change, to ensure agricultural development and food security."

Minor comments:

-L499-509: This paragraphs contains repetitions of already stated concepts. Perhaps it could be shortened.

Response: Corrected.

-L586: Please rephrase the sentence.

Response: Corrected.

6. Conclusions

- The conclusions are clear and well-balanced. However, I would recommend clearly stating the future steps to fill the existing gaps.

Response: Thank you for your observation. We have presented the future research in the discussion.

---

## Author Response (AR2)

**Response to reviewer's comments**

Editor

Dear Authors,

The paper is now ready for publication after some technical corrections: please check the comments provided by Reviewer #2. From my side, I suggest adding the word "Vietnam" to the title and slightly adjusting it to "Predicting Soil Salinity in the Red River Delta (Vietnam) Using Machine Learning and Assessing Farmers' Adaptive Capacity"; the same in the abstract, since reading it, it is difficult to understand the location of the study area.

**Responses:**

Thank you very much for your observation. We have revised the manuscript accordingly, adding "Vietnam" to the title and abstract as suggested.

**Reviewer: 1**

1. You have added a well-written section explaining how the five FAO salinity levels were selected, along with a discussion of the prevalence of relevant land use (in particular rice and aquaculture) in the corresponding study areas. While this section provides greater clarity and improves the interpretation of the results, it reads as an ex-post explanation rather than a description of a methodological step. For this reason, I suggest moving this content to the Materials and Methods section.

**Responses:**

Thank you very much for your observation. We have moved this setences in the material and method section as suggested in line 161-167.

2. In the Results section, it could then be briefly referenced, allowing you to proceed directly to the presentation of the findings (i.e., the prevalence of different crops or at least rice within each salinity class) without repeating the full explanation.

**Response:**

Thank you very much for this suggestion. We fully understand the concern about repetition; however, we decided to retain the explanation in the Results section because we believe it provides important context for readers to better interpret the findings, especially regarding the prevalence of rice and other crops in relation to different salinity classes. Many readers may consult the Results section directly without referring back to the earlier parts of the manuscript, so having this explanation available ensures clarity and coherence.